# Low rank adaptation of chemical foundation models generates effective odorant representations

**Grant D. McConachie**
Department of Biomedical Engineering
Boston University
Boston, MA 02215, USA
gdmac@bu.edu

**Emily Duniec**
Program in Neuroscience
Boston University
Boston, MA 02215
eduniec@bu.edu

**Florence Guerina**
Department of Biology
Boston University
Boston, MA 02215
fguerina@bu.edu

**Meg A. Younger**
Department of Biology
Boston University
Boston, MA 02215
myounger@bu.edu

**Brian DePasquale**[*]
Department of Biomedical Engineering
Boston University
Boston, MA 02215
bddepasq@bu.edu

## Abstract

Featurizing odorants to enable robust prediction of their properties is difficult due to the complex activation patterns that odorants evoke in the olfactory system. Structurally similar odorants can elicit distinct activation patterns in both the sensory periphery (i.e., at the receptor level) and downstream brain circuits (i.e., at a perceptual level). Despite efforts to design odorant features to better predict how they interact with the olfactory system, there is still no universally accepted approach to this problem. We demonstrate that feature-based approaches that rely on pre-trained foundation models to generate odorant representations *do not* significantly outperform classical hand-designed features on odorant-receptor binding tasks. Instead, we show that it is necessary to fine-tune these features to increase predictive performance. To show this, we introduce a new model that creates olfaction-specific representations: **L**oRA-based **O**dorant-**R**eceptor **A**ffinity prediction with **CROSS**-attention (**LORAX**). We compare existing chemical foundation model representations to hand-designed physicochemical descriptors using feature-based methods and identify large information overlap between these representations, highlighting the necessity of fine-tuning to generate novel and superior odorant representations. We show that LORAX produces a feature space more closely aligned with olfactory neural representation, enabling it to outperform existing models on predictive tasks.

## 1 Introduction

A pervasive question in olfaction is: 'What is the best way to represent odorants?' In other words, what are the best features to represent an odorant that are predictive of how that odorant will interact with the olfactory system? The answer to this question has remained elusive due to complex structure-odor relationships, where similar odorants can elicit divergent neural responses (Sell, 2006). Early efforts inspired by chemoinformatics focused on sets of hand-selected physicochemical descriptors (molecular weight, ring count, etc.) to best capture trends in neural data (Schmuker et al., 2007; Haddad et al., 2008a; Haddad et al., 2008b; Boyle et al., 2013; Gabler et al., 2013), and these descriptors remain a central tool in olfactory neuroscience (Pashkovski et al., 2020; Yang et al., 2023). More recently, deep learning has enabled the extraction of data-driven odorant features that have facilitated greater predictive accuracy and a more comprehensive mapping of odor space (Seshadri et al., 2022; Lee et al., 2023; Shuvaev et al., 2024).

---

[*]Corresponding author

Two broad approaches to data-driven odorant featurization have emerged: (1) a *supervised approach* (Figure 1B, left) where tailored architectures, such as graph neural networks (GNNs), are used to generate representations, and (2) a *feature-based approach* (Figure 1A and B, center) where chemical foundation models, which are self-supervised models that leverage unlabeled data to generate useful representations for post-hoc application, are used to generate fixed features for odorants (Shin et al., 2024; Taleb et al., 2024). The latter approach offers a promising solution to the challenges of constructing representations, particularly in olfaction where datasets are typically small. Moreover, given the estimated billions of potential odorants (Mayhew et al., 2022), self-supervised approaches may be able to effectively leverage large unlabeled molecular datasets to create more robust features for odorants not otherwise discernible by other methods.

Despite the potential utility of foundation model representations, their application to olfaction has been limited and not quantitatively and exhaustively compared to other approaches. In this work, we evaluate several chemical foundation models and find they *do not* improve predictive performance over classical physicochemical descriptors on odorant-receptor binding tasks using a solely feature-based approach (Section 3). Using two statistical shape analysis methods, canonical correlation analysis (CCA) and orthogonal Procrustes analysis, we find many of these representations have strong information overlap with physicochemical features, suggesting that self-supervised learning alone does not yield features well suited for odorant-receptor binding prediction.

To address this limitation, we introduce a model that uses a *fine-tuning based approach* (Figure 1B, right) to produce novel odorant representations tailored to olfaction. Our model, **L**oRA-based **O**dorant-**R**ecptor **A**ffinity prediction with **CROSS**-attention (**LORAX**; Figure 3), employs low-rank adaptations (Hu et al., 2021) of chemical foundation models to fine-tune their representations over training (Section 4). We demonstrate LORAX's learned odorant representation allows for better performance on odorant-receptor binding tasks and better generalization. Furthermore, we show the learned LORAX representation is more aligned with neural responses, providing insight into the cause of increased model performance.

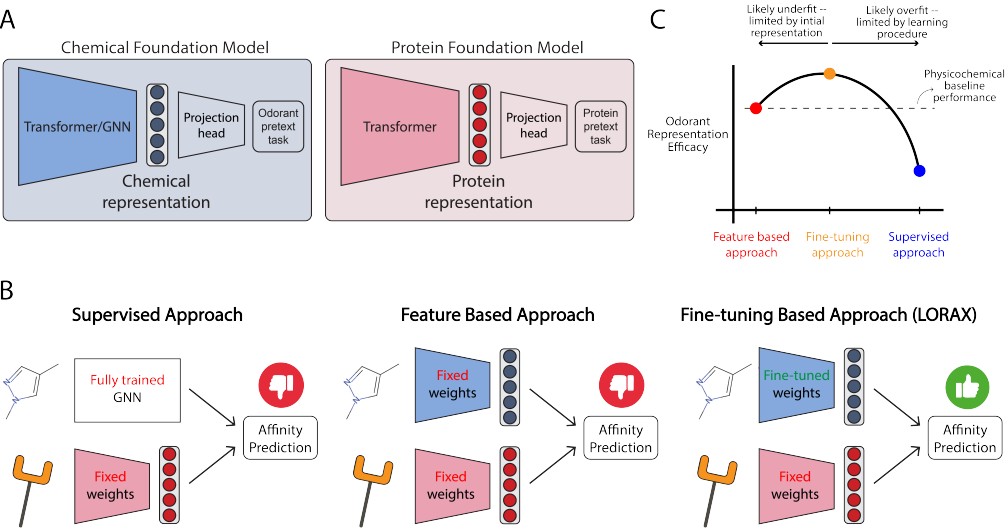

Figure 1: LORAX presents a new approach to incorporate chemical foundation models to predict odorant-receptor affinity. (A) General architecture of chemical and protein foundation models. (B) Current approaches, and our proposed approach (right), to predicting affinity of odorant-receptor binding. (C) Putative thought model given our approach and results: we hypothesize that fine-tuning allows for creation of improved odorant representations that outperform existing approaches.

## 2 RELATED WORKS

**Physicochemical features to predict affinity.** Guo & Kim (2010) predict odorant-receptor relationships using partial least squares by representing odorants with GRIND descriptors (Pastor et al., 2000) and encoding receptors with multidimensional scaling. Boyle et al. (2013) and Gabler et al. (2013) use optimized physicochemical descriptors to predict receptor activation, employing hierarchical clustering, and support vector machines and random forests, respectively. Cong et al. (2022) uses physicochemical descriptors of odorants and multiple sequence alignment similarities of receptors to train a random forest that generates predictions. Berwal et al. (2025) uses molecular docking simulations with K-means clustering of odorants based on physicochemical descriptors to generate affinity predictions.

**Supervised approach to predict affinity.** Achebouche et al. (2022) develop graph and convolutional neural networks to predict receptor activation. Hladiš et al. (2022) and Chithrananda et al. (2024) use GNNs to represent odorants combined with a protein foundation model to predict odorant-receptor binding. None of these approaches investigate chemical foundation models as odorant featurizers: while Chithrananda et al. (2024) and Hladiš et al. (2022) incorporate *protein* foundation models into odorant-receptor affinity prediction, they employ randomly initialized GNN architectures with supervised training rather than pre-trained models.

**Pre-trained and fine-tuned chemical foundation models in olfaction.** Shin et al. (2024) use the chemical foundation models MolCLR (Wang et al., 2022) and SMILES Transformer (Honda et al., 2019) to represent odorants to predict odor percepts. Similarly, Taleb et al. (2024) use MoLformer (Ross et al., 2022) to featurize odorants and predict their percept. Both groups found chemical foundation models offer valuable features that can aid odor prediction. However, these groups do not test chemical foundation models to predict odorant-receptor affinity and do not utilize as broad a selection of chemical foundation models as we do in this work. To our knowledge, chemical foundation models have not been applied to predict odorant-receptor interactions, and different chemical foundation models have not been systematically benchmarked to evaluate their effectiveness. Furthermore, the LoRA-based fine tuning we introduce and the performance improvements it produces have not been documented for olfactory tasks.

## 3 DO CHEMICAL FOUNDATION MODELS CREATE EFFECTIVE FEATURES?

We benchmarked a variety of chemical foundation models using three feature-based approaches to examine which representations, if any, enhance odorant-receptor affinity prediction. To do this, we compiled four datasets highlighted in the section below.

### 3.1 DATASETS

Each dataset includes the amino acid sequence of the receptor, a chemical identifier of the odorant (e.g., SMILES string (Weininger, 1988)), and an experimental measurement of odorant-receptor affinity (summary of datasets in Table 12). These datasets span mammals and non-mammals and employ diverse methodologies for assessing odorant-receptor interactions.

**Hallem Dataset.** Hallem & Carlson (2006) reports the responses of 24 *D. melanogaster* receptors to 110 odorants using electrophysiology, yielding a total of 2,640 odorant-receptor pairs. Responses are average spikes per second. We normalize the data by $z$-scoring all responses, using the mean and standard deviation of the entire dataset (i.e., we do not $z$-score responses for each receptor individually).

**Carey Dataset.** Carey et al. (2010) reports the responses of 50 *A. gambiae* receptors to 110 odorants using electrophysiology, yielding a total of 5,500 odorant-receptor pairs. Responses are quantified as in the Hallem dataset. We normalize the dataset by $z$-scoring all responses, as in the Hallem dataset.

**M2OR.** Mammalian odorant-receptor interactions collected from 45 different sources (Lalis et al., 2024). The dataset contains 771 odorants and 1402 receptors across 16 species totaling 53,444 paired responses. We used a subset of 46,563 odorant-receptor pairs to directly compare LORAX to previous models trained on this dataset (Section 4). Each paired response is categorized as either

one for responsive or zero for non-responsive. We do not include this full dataset in our feature-based model benchmarks (Section 3.3); for those analyses we instead use the higher-quality M2OR (EC50) subset described below.

**M2OR (EC50).** The M2OR dataset has a variety of data quality. Much of the data comes from 'primary' or 'secondary' screening, which are not as precise as data collected in the Hallem and Carey datasets. For Section 3.3, we use the M2OR 'EC50' data, as it is the highest quality. This subset of the data contains 474 odorants and 503 receptors, yielding a total of 5,834 odorant-receptor pairs.

## 3.2 BENCHMARKING MODELS

To effectively benchmark a variety of chemical foundation models against standard methods we use fifteen different representations: five transformer-based foundation models, three GNN-based foundation models, eight physicochemical descriptors, and a randomly generated representation as a lower bound (see Appendix A for more details). We use three feature-based models (i.e., models that are not fine-tuned) to benchmark these representations, highlighted below. For models that incorporate a protein foundation model, we use ESM (Rives et al., 2021). As we are predominantly interested in odorant representations, we used only one protein foundation model.

**Molecule only model (MO).** A ridge regression model using only odorant representations. This model is trained on every receptor individually[1] and results are averaged over all receptors. This model assesses the effectiveness of the odorant representations alone, testing whether only chemical information can predict receptor activation.

**Molecule + protein model (MP).** A ridge regression model[2] that uses an odorant representation concatenated with a protein representation to predict affinity. This model tests the ability of the odorant representation to predict binding with additional receptor target information.

**ProSmith.** We adapt a state-of-the-art multi-modal transformer (Kroll et al., 2024) to test odorant representations for affinity prediction. This model takes, as input, a chemical and protein representation and passes them through a transformer and a gradient boosted decision tree (XGBoost, (Chen & Guestrin, 2016)) ensemble to generate predictions. Importantly, this model does not fine-tune either representation during training.

Further details of these benchmarking models are provided in Appendix B and C. Performance was evaluated using 5-fold cross-validation with random splits[3]. Averaged prediction metrics are reported for the Carey, Hallem, and M2OR (EC50) datasets.

## 3.3 CHEMICAL FOUNDATION MODELS DO NOT IMPROVE ODORANT-RECEPTOR AFFINITY PREDICTION

Benchmarking results are shown in Table 1 and Appendix G. We focus on the Carey dataset in the main text, but the observations and conclusions are consistent across the M2OR (EC50) and Hallem datasets as well. From these results, we draw three conclusions. First, molecular information alone is insufficient to do prediction. This conclusion is supported by the MO column in Table 1. We see low $R^2$ scores with high variance across folds, indicating that solely using the odorant representations is insufficient for accurate prediction. Both physicochemical and chemical foundation model representations lack sufficient contextual information to predict binding.

Second, incorporating protein information is essential for achieving reliable prediction. This can be seen by the large performance increase from the MO to MP models, showing that incorporating receptor information increases $R^2$ scores and decreases variance across folds. Additionally, we obtain very high predictive power using the ProSmith model when both receptor and odorant representations are combined through a transformer. This highlights the importance of multi-modal approaches in this domain.

---

[1]For the M2OR (EC50) dataset, we exclude receptors with fewer than 50 entries when training the MO model, as this amount of data is too limited to ensure reliable training.

[2]The MP and MO models are a logistic regression with an $\ell_2$ penalty for the M2OR (EC50) dataset, as responses are binary.

[3]We use a random stratified split for the MO model on the M2OR (EC50) dataset.

Table 1: Affinity prediction performance across odorant representations for the Carey dataset for the three models outlined in Section 3.2. Reported as mean coefficient of determination ($R^2$) $\pm$ standard deviation across 5-fold cross validation. Representation type depicted in the leftmost column. Best performing representations are bolded.

| | Odorant representation | MO | MP | ProSmith |
|---|---|---|---|---|
| **Physicochemical** | CATS | $0.110 \pm 0.227$ | $0.271 \pm 0.013$ | $0.637 \pm 0.019$ |
| | MordredDescriptors | $0.178 \pm 0.261$ | $0.295 \pm 0.022$ | $0.719 \pm 0.021$ |
| | Pharmacophore2D | $0.175 \pm 0.261$ | $0.282 \pm 0.018$ | $0.548 \pm 0.014$ |
| | RDKitDescriptors2D | $0.034 \pm 0.404$ | $0.264 \pm 0.021$ | $\mathbf{0.720 \pm 0.026}$ |
| | ScaffoldKeyCalculator | $0.005 \pm 0.280$ | $0.246 \pm 0.016$ | $0.594 \pm 0.022$ |
| | ECFP | $0.207 \pm 0.410$ | $0.297 \pm 0.020$ | $0.664 \pm 0.034$ |
| **Transformer** | Roberta-Zinc480M-102M | $0.181 \pm 0.237$ | $0.295 \pm 0.020$ | $0.672 \pm 0.030$ |
| | GPT2-Zinc480M-87M | $0.121 \pm 0.266$ | $0.296 \pm 0.023$ | $0.668 \pm 0.027$ |
| | ChemGPT-19M | $-0.029 \pm 1.046$ | $0.285 \pm 0.028$ | $0.685 \pm 0.024$ |
| | MolT5 | $0.215 \pm 0.279$ | $0.302 \pm 0.018$ | $0.676 \pm 0.022$ |
| | ChemBERTa-77M-MTR | $0.218 \pm 0.298$ | $0.290 \pm 0.019$ | $0.704 \pm 0.017$ |
| **GNN** | gin_supervised_infomax | $0.243 \pm 0.273$ | $\mathbf{0.303 \pm 0.021}$ | $0.671 \pm 0.016$ |
| | gin_supervised_edgepred | $0.189 \pm 0.308$ | $0.295 \pm 0.019$ | $0.687 \pm 0.027$ |
| | gin_supervised_contextpred | $\mathbf{0.257 \pm 0.255}$ | $\mathbf{0.303 \pm 0.021}$ | $0.699 \pm 0.033$ |
| | random | $-0.071 \pm 0.137$ | $0.294 \pm 0.023$ | $0.586 \pm 0.037$ |

Third, the choice of odorant representation has little effect on predictive performance. We performed a one-way ANOVA followed by a Bonferroni-corrected post-hoc Tukey test on our ProSmith results. Our analysis showed, with the exception of CATS, Pharmacophore2D, and ScaffoldKeyCalculator, all representations performed equivalently. This is surprising, especially given the success of foundation models in other domains. One would expect that chemical foundation model representations would provide richer molecular context for predicting odorant-receptor affinity, but given results highlighted here, it seems these representations do not offer important features that enhance predictability.

### 3.4 CHEMICAL FOUNDATION MODEL REPRESENTATIONS ARE ALIGNED WITH PHYSICOCHEMICAL FEATURES

To understand the failure of foundation models to improve prediction, we analyzed foundation model odorant representations by calculating dissimilarity metrics between each representation (Williams et al., 2022). We embedded all odorants in the Carey dataset into their respective feature spaces and calculated both orthogonal Procrustes distance metrics (Figure 2a) and CCA distance metrics (Figure 2b) between each pair of representations (see Appendix D for more details).

From these distance matrices, particularly the Procrustes matrix, we see that many of the odor spaces are aligned. The only two representations that are visibly dissimilar (ScaffoldKeyCalculator and Pharmacophore2D) are the worst performing representations (Table 1), while every other representation forms a block of small distances in the upper left. The odorant representations that are more similar also share similar performance in Table 1, for example RDKitDescriptors2D and Mordred-Descriptors have a very low Procrustes distance of 0.61 and perform almost identically when used in ProSmith. In addition to performance trends, this analysis uncovers other patterns present in the representations. For example, the most similar representations, gin_supervised_contextpred and gin_supervised_edgepred, are generated from two instantiations of the same GNN (Hu et al., 2020). We also see that the ChemBERTa-77M-MTR representation, a foundation model trained on 77 million SMILES strings to predict RDKit descriptors, is similar to the RDKitDescriptors2D representation. Analyzing these spaces, and their dissimilarities, explains why many of these descriptors perform the same: there is heavy overlap of information content, making the representations fairly homogeneous and redundant.

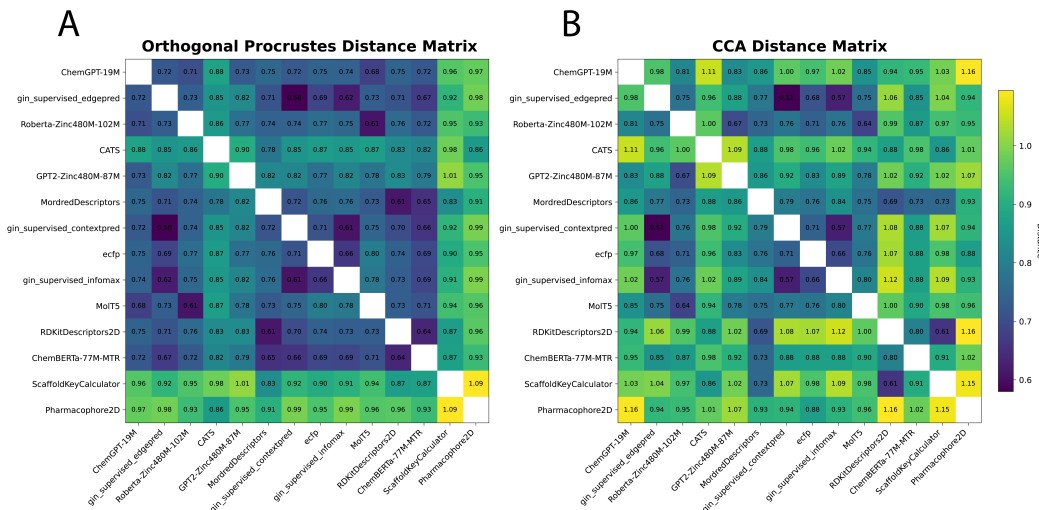

Figure 2: Distance metrics between odorant representations in the Carey dataset. Shown are (A) orthogonal Procrustes distance and (B) CCA distance metrics. See Williams et al. (2022) and Appendix D for details on these metrics.

## 4    LORAX: FINE TUNING CHEMICAL FOUNDATION MODELS WITH LoRA

Given the observed redundancy among chemical representations, we developed a model that is able to fine-tune, and adapt, these chemical foundation models to create an *olfaction-tailored representation*. All models tested in Section 3 are feature-based, relying on fixed chemical (and protein) representations rather than adapting them through fine-tuning. A key advantage of fine-tuning is that it produces a new, task-tailored representation rather than depending solely on pre-trained embeddings. Prior work has demonstrated that fine-tuning is generally more effective than feature-based methods (Devlin et al., 2019), and adapter techniques, such as LoRA (Hu et al., 2021), enable efficient fine-tuning of large models. Given that many of the foundation model representations have large information overlap, we predicted that refining these representations would yield more informative and task relevant features for olfactory tasks. Motivated by these considerations, we introduce LORAX (Figure 3).

LORAX is a multi-modal transformer that incorporates a protein and a chemical foundation model to predict affinity (Figure 3a, 8). LORAX uses LoRA to update the chemical representation over training. The protein and chemical foundation model can be any model on Huggingface, but for all analysis in this section, we use ESM (Rives et al., 2021) to represent receptors and ChemBERTa-77M-MTR (Ahmad et al., 2022) to represent odorants[4]. We take inspiration from Kroll et al. (2024) and incorporate an ensemble XGBoost models to make final predictions (Figure 3b). Each XGBoost model in the ensemble uses a distinct set of input features: (1) the transformer's penultimate representation (referred to as the `<cls>` token), (2) the concatenated original chemical and protein foundation model representations, and (3) the concatenation of the `<cls>` token with the original chemical and protein foundation model representations. Training proceeds in two stages. First, the multi-modal transformer (including the LoRA parameters) is trained and tuned on the training and validation data, refining the `<cls>` token representation. After this stage, the transformer is frozen and the resulting `<cls>` token, together with the fixed foundation-model embeddings, are used as input features to train the XGBoost ensemble, which makes the final predictions on the held-out test data. Architecture and training details are available in Appendix F.

---

[4]We also test LORAX with Uni-Mol2-84M (Ji et al., 2024) as the chemical foundation model and show these data in supplementary Table S8.

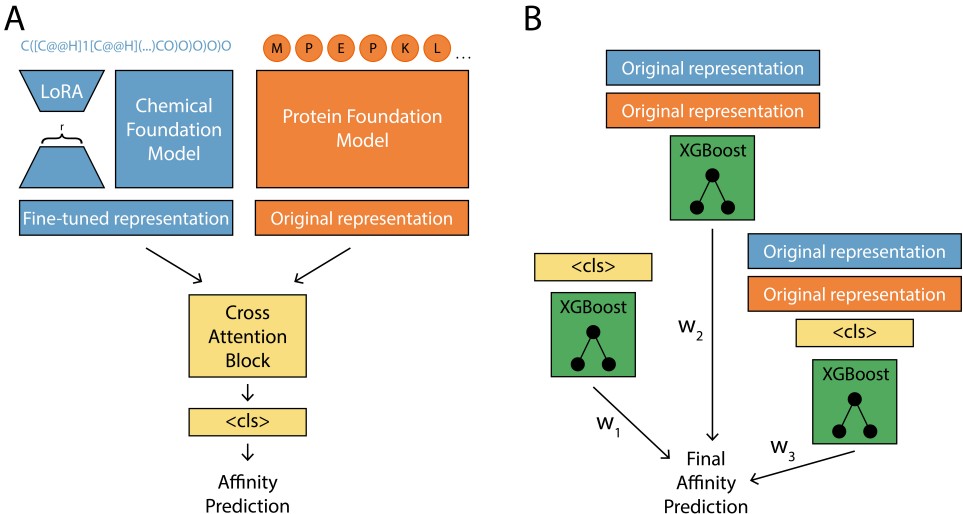

Figure 3: LORAX model. (A) A multi-modal LoRA adapted transformer model with a cross attention block for odorant-receptor prediction. $r$ is the rank of the low rank adapter matrices. (B) An ensemble XGBoost models are used to enhance performance while also providing model interpretability. A weighted sum of these XGBoost models is used to generate the final prediction. 'Original representation' means the non-LoRA fine-tuned foundation model representation. Orange represents the protein foundation model and blue represents the chemical foundation model.

### 4.1 LORAX CONSTRUCTS A BETTER ODORANT REPRESENTATION

We trained LORAX on the Carey dataset and found it performs almost identically to ProSmith (LORAX $R^2$: 0.712 $\pm$ 0.032, ProSmith with ChemBERTa-77M-MTR $R^2$: 0.703 $\pm$ 0.017, p=0.264 paired t-test, data not shown). However, when we interrogated the trained models, we uncovered nuanced differences between them. Because ProSmith and LORAX have similar architectures and both employ identical XGBoost ensembles, a direct comparison between them is possible. When examining the ensemble weights (Figure 4B, S1), we found that ProSmith places high confidence in the XGBoost model that uses only the original chemical and protein foundation model representations. In other words, the additional representation produced by the transformer (the <cls> token) is largely ignored. This is supported by the relatively weak standalone validation accuracy of the multimodal transformer model, prior to the XGBoost ensemble, (Figure 4A, gray) showing ProSmith's transformer lacks predictive performance. When we compute these metrics for LORAX (Figure 4, blue), we see the validation accuracy of its multi-modal transformer is much higher and the weights placed on XGBoost models without the <cls> token are lower. We conclude that LORAX extracts more relevant information for the task from the protein and chemical foundation models.

With this context, we next examined whether the differences in representation weighting between ProSmith and LORAX translate into improved odorant representations across different scenarios. We first wanted to assess the generalizability of LORAX. Using the Carey dataset, we trained both LORAX and ProSmith[5] on two scenarios: (1) generalize to unseen receptors (i.e., receptors in the test set are never seen in training), and (2) generalize to unseen odorants (i.e., odorants in the test set never appear in training). As shown in Table 2, LORAX exhibits superior generalization to unseen odorants over ProSmith. Although this improvement is not statistically significant (p=0.069, BH-corrected Friedman test (Table S5)), it indicates that LORAX representations may capture more chemically relevant features for odorant–receptor binding prediction. In the unseen receptor scenario, LORAX and ProSmith perform equivalently (p=1.000, (Table S6)). Overall, while generalization for both models remains challenging, LORAX notably outperforms the naïve baseline, a

---

[5]We train ProSmith with the ChemBERTa-77M-MTR model for appropriate comparison.

model predicting the mean value of the training set, in both scenarios (p=0.069 (Tables S5 and S6)) while ProSmith fails to outperform the naïve model on the unseen odorant task (p=1.000 (Table S5)).

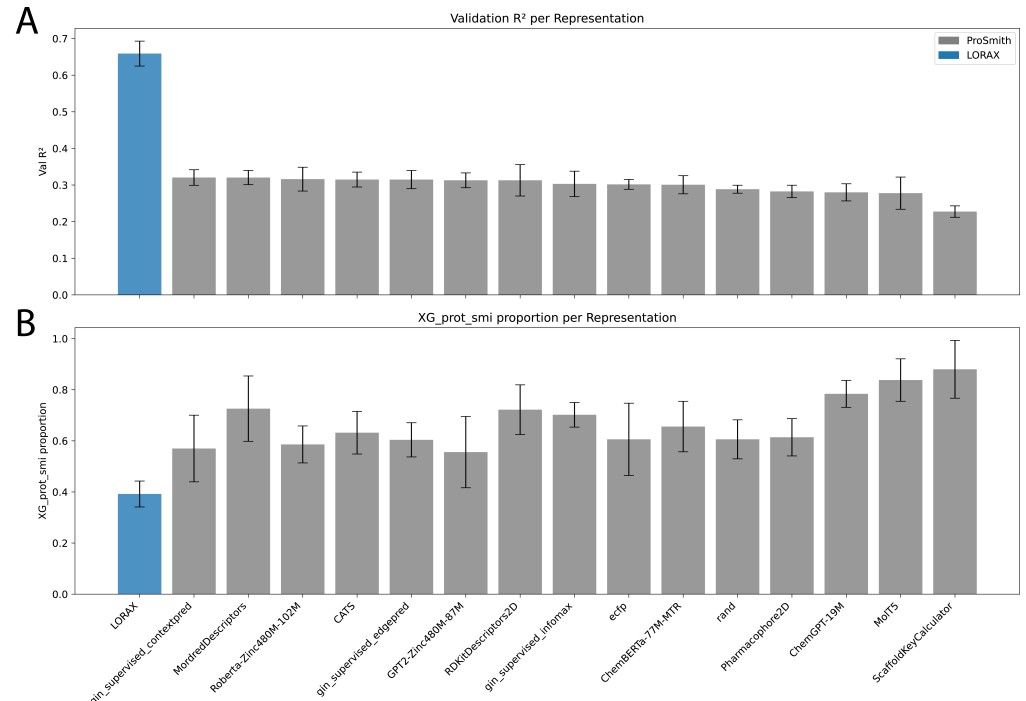

Figure 4: LORAX and ProSmith representation comparison after training on the Carey dataset. (A) Validation $R^2$ scores for ProSmith and LORAX prior to applying XGBoost. (B) The weight assigned to the XGBoost model that takes only the original chemical foundation model and protein foundation model representations.

Table 2: LORAX, ProSmith, and naïve model performance on the Carey dataset under two generalization settings. The naïve model predicts the mean of the training set for all values in the test set. Shown are $R^2$ scores for the test set for each random split. Bolded are the best performing models.

| Scenario | Method | split_1 | split_2 | split_3 | split_4 | split_5 | avg | std |
|---|---|---|---|---|---|---|---|---|
| Unseen odorants | ProSmith | -2.098 | -0.812 | 0.364 | 0.246 | 0.288 | -0.402 | 1.064 |
| | LORAX | -1.039 | 0.201 | 0.472 | 0.292 | 0.298 | **0.045** | 0.614 |
| | Naïve | -0.037 | -0.051 | -4.924 | -0.050 | -0.020 | -1.016 | 1.954 |
| Unseen receptors | ProSmith | 0.072 | 0.246 | 0.263 | -0.137 | -0.003 | **0.088** | 0.169 |
| | LORAX | -0.032 | 0.051 | 0.031 | 0.133 | 0.117 | 0.060 | 0.067 |
| | Naïve | -0.060 | -0.096 | -0.019 | -0.129 | -0.059 | -0.072 | 0.037 |

To further interrogate LORAX, we examine how the model performs relative to other models on the full M2OR dataset, a larger dataset that presents greater diversity and complexity, offering a more rigorous test of model performance. We utilize the weighting scheme outlined in Hladiš et al. (2022) to account for differing data quality. As shown in Table 3, LORAX demonstrates superior performance compared to both Hladiš et. al. and MolOR, yielding a statistically significant improvement over the latter (p<0.003, MCC across splits[6] (Table S7)). LORAX also outperforms ProSmith, which is evaluated here on M2OR for the first time (p=0.067, MCC across splits (Table S7)). While

---

[6]A statistical comparison against the model from Hladiš et al. (2022) could not be computed due to lack of access to per-split metrics.

ProSmith achieves a higher AUROC, LORAX demonstrates improvements in MCC and F-score over state-of-the-art protein-molecule interaction models not previously tested on M2OR. Given the large class imbalance in the M2OR dataset, where the proportion of responsive data points is much lower than that of non-responsive ones (Table S1), MCC and F-score provide a more accurate assessment of model performance than AUROC. This confirms that fine-tuning chemical foundation models offers an important avenue to improve odorant-receptor prediction.

Table 3: Comparison of MolOR, Hladiš et al., ProSmith, and LORAX across multiple metrics trained on the M2OR dataset. LORAX (C) fine-tunes only the chemical foundation model; LORAX (P + C) fine-tunes both chemical and protein foundation models. Values for Hladiš et al. are taken from their publication. Uncertainties for all models are standard deviation across 5 fold cross validation. Bold indicates highest value in each column. n/a entries indicate that the authors did not report that metric. AUROC: area under the receiver operating characteristic curve, AveP: average precision, MCC: Matthews correlation coefficient.

|  | AUROC | AveP | Precision | Recall | F-score | MCC |
|---|---|---|---|---|---|---|
| MolOR (Weighted) | $76.12 \pm 1.53$ | $0.626 \pm 0.05$ | $0.507 \pm 0.05$ | $0.727 \pm 0.06$ | $0.595 \pm 0.02$ | $0.467 \pm 0.03$ |
| Hladiš et. al. | n/a | $\mathbf{0.780 \pm 0.01}$ | $0.689 \pm 0.02$ | $0.698 \pm 0.04$ | $0.693 \pm 0.02$ | $0.605 \pm 0.02$ |
| ProSmith | $\mathbf{90.47 \pm 0.98}$ | $0.584 \pm 0.03$ | $\mathbf{0.764 \pm 0.06}$ | $0.671 \pm 0.06$ | $0.712 \pm 0.02$ | $0.641 \pm 0.03$ |
| LORAX (P + C) | $82.43 \pm 0.86$ | $0.776 \pm 0.03$ | $0.729 \pm 0.06$ | $0.727 \pm 0.03$ | $0.727 \pm 0.02$ | $0.650 \pm 0.03$ |
| LORAX (C) | $83.24 \pm 1.34$ | $0.778 \pm 0.03$ | $0.710 \pm 0.05$ | $\mathbf{0.754 \pm 0.02}$ | $\mathbf{0.730 \pm 0.03}$ | $\mathbf{0.651 \pm 0.04}$ |

We also tested a version of LORAX that uses both a LoRA adapted protein foundation model and a LoRA adapted chemical foundation model (LORAX (P + C)). Interestingly, this model performs slightly worse than the model with only chemical foundation model fine-tuning, even though LORAX (P + C) has ~2 million more parameters than LORAX (C). We conclude that the task benefits mainly from LORAX's improved chemical representations, while the original protein representations are already sufficient. In summary, we found that LORAX fine-tuning produced models with more refined `<cls>` tokens, translating to improved prediction on M2OR and improved generalizability to unseen odorants.

## 4.2 LORAX ODOR SPACE IS SIMILAR TO UNDERLYING NEURAL REPRESENTATION

Given LORAX's promising capabilities, we analyzed the odorant representations it produces. We plot CCA distances of LORAX's odor space for the Carey dataset in Figure 5.

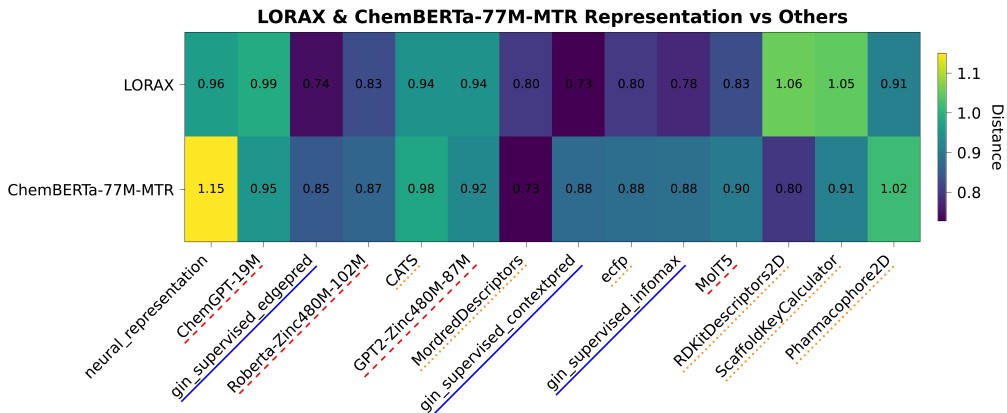

Figure 5: CCA distance matrix showing the LORAX and ChemBERTa-77M-MTR representations on the Carey dataset. 'neural_representation' encodes each odorant as a feature vector based on the responses it elicits from each neuron. Red dashed lines indicate transformer-based foundation model representations, blue solid lines indicate GNN-based foundation model representations, and orange dotted lines indicate physicochemical representations. Full distance matrices shown in Figure 7.

Additionally, for this distance matrix, we include a 'neural_representation', in which each odorant is represented as a vector of the neural responses it elicits (more details in Appendix E). We compare the LORAX representation with ChemBERTa-77M-MTR, the foundation model that is adapted using LoRA during training, for the Carey dataset. The ChemBERTa-77M-MTR representation can be considered the initial state before fine-tuning while the LORAX representation represents the final state after adaptation.

We found the LORAX representation is more dissimilar from many of the physicochemical descriptors than ChemBERTa-77M-MTR and more similar to all graph based methods (ECFP and all the GNN foundation models). Additionally, the LORAX representation is more aligned with the neural representation than the ChemBERTa-77M-MTR representation. These analyses reveal that LORAX is generating a unique representation, distinct from the original foundation model, that is better suited to describe odorant-receptor relationships and therefore generating a more informative odor space.

## 5 DISCUSSION

Identifying representations that best characterize odorants for olfactory prediction has been a major research focus. We systematically evaluated representations from several pre-trained chemical foundation models and found they did not provide an improvement over hand-tuned physicochemical descriptors for odorant-receptor binding tasks using multiple feature-based approaches. To move beyond this limitation, we introduced a new model, LORAX, for fine-tuning chemical foundation models to produce richer and more powerful representations. Collectively, we present two novel and interconnected findings: **(1)** To our knowledge, we performed the first and most comprehensive analysis of chemical foundation models applied to odorant-receptor binding to demonstrate their effective equivalence with physicochemical descriptors; **(2)** We introduced the first application of LoRA fine-tuning of chemical foundation models for olfaction, demonstrating the improvement that fine-tuning offers. Our results demonstrate that while existing chemical foundation models may capture overlapping information for olfactory datasets, targeted fine-tuning creates specialized and potentially more powerful representations than those obtained from pre-trained models alone. We argue that these models have significant potential for olfactory neuroscience, particularly in addressing key challenges such as limited data and vast chemical search spaces.

We hypothesize the targeted fine-tuning LoRA provides identifies a 'sweet spot' that balances the strengths of alternative approaches (Figure 1C). Supervised approaches, that lack the context of pre-trained models, underperform likely due to overfitting (Hladiš et. al., Table 3). Feature-based models alone likewise underperform because a limited initial representation leads to underfitting (ProSmith, Table 3). We hypothesize that the selective nature of LoRA's low-rank assumption enables robust generalization while not entirely erasing the benefits that come from pre-training. To the best of our knowledge, this work presents the first application of LoRA to olfaction, and given its strength and natural approach to balancing over- and underfitting when fine-tuning pre-trained models, we expect it to be a useful tool when working with olfactory data.

There remain many exciting avenues for future exploration with this modeling framework. A dominant question in olfaction is how chemical features produce olfactory perception (Zhou et al., 2018; Seshadri et al., 2022; Lee et al., 2023; Yang et al., 2023; Qian et al., 2023; Shuvaev et al., 2024). Shin et al. (2024) and Taleb et al. (2024) found that pre-trained models with non parameter-efficient or no fine-tuning improved performance on perception tasks; our results suggest further improvement could be achieved with LoRA-based fine-tuning. Additionally, LORAX is modular by design, allowing for the interchange of protein and chemical foundation models. While our analysis utilizes ChemBERTa-77M-MTR (Ahmad et al., 2022), the field of chemical foundation modeling is rapidly advancing with larger and more diverse models. Systematic benchmarking and further training with these new models may yield even more optimal odorant representations.

Despite our progress, several important questions remain. For example, how robust are the LORAX-learned representations when transferring to different olfactory datasets? Which components of LORAX contribute the most to prediction performance? Why does simultaneous fine-tuning of both protein and chemical foundation models hamper performance? We leave these to future work, with the anticipation that continued research will deepen our understanding of how chemical foundation models can be effectively leveraged in olfaction.

## REPRODUCIBILITY STATEMENT

All datasets used in this work are freely available and from published work linked in the text. Code of the benchmarking work in Section 3 is located here, and code for the LORAX model is located here. Data used in the work can be downloaded from links in either README. Default configurations and training regimes are outlined in the main text and appendices for all models used in this work. Statistical shape analysis methods in this work are freely available to use here.

## USE OF LLMS IN THIS WORK

We use LLMs to aid and polish writing throughout the text.

## ACKNOWLEDGMENTS

This material is based upon work supported by the National Science Foundation Graduate Research Fellowship Program under Grant No 2234657. Any opinions, findings, and conclusions or recommendations expressed in this material are those of the authors and do not necessarily reflect the views of the National Science Foundation. GDM acknowledges NIH T32-GM008764.

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

## A  DATA REPRESENTATION

Here we describe the various methods we use to featurize odorants and receptors. For odorant featurization, we wanted a diverse array of chemical foundation models as well as physicochemical descriptors to get a good sense of what features need to go into an odorant representation to be able to effectivity predict how that odorant binds with a given receptor. The odorant representations we used are described in Table 4. For all odorant representations in Section 3 we use Molfeat (Noutahi et al., 2023) to generate representations. In Section 4, we use Huggingface to generate molecular representations as we focus on finetuning the original models when we train LORAX.

We would also like to note that the generation of chemical foundation models is a rapidly progressing field, and we realize that some of the most recent chemical foundation models are not present in this study. However, we would like to highlight that we cover a variety of architectures, pre-training tasks, and model sizes in our selection of chemical foundation models.

As we are mainly interested in exploring the efficacy of different odorant representations, we only use a single protein foundation model ESM (Rives et al., 2021). For benchmarking in Section 3 we use 'esm1b_t33_650M_UR50S', the ESM model used by default in ProSmith (Kroll et al., 2024). In Section 4, we use a newer version of the ESM model 'esm2_t33_650M_UR50D'.

Table 4: Summary of molecular descriptors used in this work.

| Descriptor | Source | Description |
|---|---|---|
| CATS | Physicochemical | A representation designed for drug discovery applications that uses a histogram of distances between atom pairs to represent a molecule (Schneider et al., 1999). We use the 2D version. |
| MordredDescriptors | Physicochemical | A collection of 1,800 features representing the physicochemical, topological, geometrical, and constitutional properties of molecules (Moriwaki et al., 2018). We remove NaN values and z-score each feature. |
| Pharmacophore2D | Physicochemical | Computes topological distances between all pairs of atoms within a molecule. Atoms are featurized with a variety of physical properties. These feature pairs and distances are encoded into a fixed-length bit vector; we use a 2048-length vector. |
| RDKitDescriptors2D | Physicochemical | All 2D physicochemical descriptors from RDKit, an open-source cheminformatics toolkit for molecular representation, manipulation, and analysis. |
| ScaffoldKeyCalculator | Physicochemical | Encoding of simple substructure features of molecular scaffolds, such as hydrogen bond donors, acceptors, and aromatic rings, into a fixed-length binary vector (Ertl, 2021). We use a 42-length vector. |
| ECFP | Physicochemical | Circular molecular fingerprints that represent a molecule by iteratively encoding the local atomic environments around a defined radius (Rogers & Hahn, 2010).We use a 2000-length vector. |
| Roberta-Zinc480M-102M | Transformer | A RoBERTa style transformer with 102M parameters trained with a masked language model objective on 480M SMILES strings (Heyer, 2023). |
| GPT2-Zinc480M-87M | Transformer | A GPT2 style transformer with 87M parameters trained on a next character prediction task on 480M SMILES strings (Heyer, 2024). |
| ChemGPT-19M | Transformer | A GPT3 style transformer with over 1B parameters trained on a next character prediction task on 10M SELFIES strings (Frey et al., 2022). |
| MolT5 | Transformer | A T5 style transformer trained on both SMILES strings and natural language text (Edwards et al., 2022). This model uses a 'replace corrupted spans' task, where the model samples natural language text and a SMILES string simultaneously, masks various parts of each, then trains the model to predict masked sections. |
| ChemBERTa-77M-MTR | Transformer | A RoBERTa style transformer trained to predict a set of 200 molecular properties on 77M SMILES strings (Ahmad et al., 2022). |
| gin_supervised_infomax | GNN | A graph isomorphism network (GIN) trained on 2M molecules to maximize the mutual information between local node representations and whole graph representations (Hu et al., 2020; Veličković et al., 2018). |
| gin_supervised_edgepred | GNN | A GIN trained on 2M molecules to predict masked edge attributes of molecular graphs (Hu et al., 2020). |
| gin_supervised_contextpred | GNN | A GIN trained on 2M molecules to use subgraphs of the molecules to predict their surrounding graph structures (Hu et al., 2020). |
| random | random | As a baseline, we generate a random vector representation for every odorant. This gives us a lower bound on performance. |

# B  PROBLEM STATEMENT AND MODEL DESCRIPTIONS FOR BENCHMARKING STUDY

We represent any featurization of an odorant as a vector $x \in \mathbb{R}^n$, where $n$ is the embedding dimension of the representation.[7] Similarly, we represent a receptor as a matrix $P \in \mathbb{R}^{l_p \times m}$, where $l_p$ is the sequence length of the receptor (number of amino acids)[8]. Given an odorant–receptor pair $\{x, P\}$, our goal is to predict the experimentally measured affinity $y$ with various odorant embeddings:

$$f(\{x, P\}) = y. \tag{1}$$

We use three different models to assess the efficacy of chemical foundation models. A schematic of all three models can be seen in Figure 6. Code for all of these models is available here.

---

[7]Some transformer models produce representations that scale with the sequence length of the molecule. To compare these fairly with physicochemical descriptors and GNN-based embeddings, we mean-pool across the sequence dimension.

[8]The receptor embedding dimension $m$ is determined by the protein foundation model.

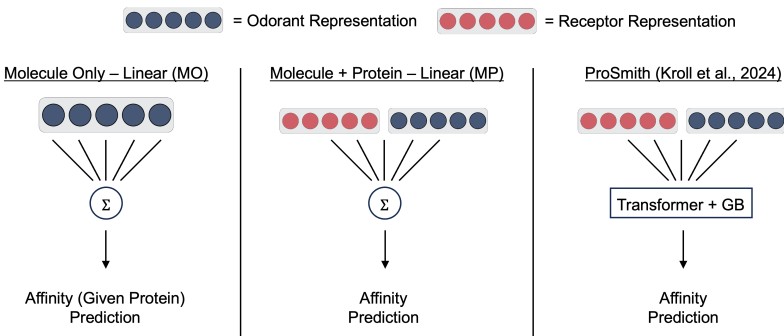

Figure 6: Schematic showing all the benchmarking models used in Section 3. Odorant representation can be any representation highlighted in Table 4. Protein representation is ESM. GB: gradient boosting

## B.1 MOLECULE ONLY MODEL (MO)

A ridge regression model that predicts affinity using only chemical foundation model representations. For each receptor, the model takes the form

$$f(\boldsymbol{x}) = y \tag{2}$$

where only the odorant representation $\boldsymbol{x}$ is used. For each receptor, we run a nested 5-fold cross validation. We conduct a grid search over regularization parameters from 21 values evenly spaced on a logarithmic scale between $10^{-10}$ to $10^{10}$. For the M2OR (EC50) dataset, as it is a binary classification task instead of a regression, we use a $\ell_2$ penalized logistic regression instead of a ridge regression. For M2OR (EC50) we use a stratified split to ensure there is both positive and negative examples in each of the splits. Additionally, we ignore proteins with $<50$ odorants, as there is not enough data associated with those odorants to ensure reliable prediction.

## B.2 MOLECULE + PROTEIN MODEL (MP)

A ridge regression model that uses both molecular and protein embeddings. Specifically, we concatenate the odorant embedding with a mean-pooled protein embedding

$$(\boldsymbol{x}||\boldsymbol{p}) \in \mathbb{R}^{n+m}, \quad \boldsymbol{p} = \mathrm{meanpool}(\boldsymbol{P}) \tag{3}$$

where $||$ denotes concatenation, and train

$$f(\boldsymbol{x}||\boldsymbol{p}) = y \tag{4}$$

This model, similarly to the MO model, uses a nested 5-fold cross validation scheme. We again use logistic regression with an $\ell_2$ penalty to train this model on M2OR (EC50). We no longer take out any data from the M2OR (EC50) dataset, as we did in the MO model.

## B.3 PROSMITH

A state-of-the-art small molecule protein multi-modal transformer model (Kroll et al., 2024). The odorant and receptor representations are projected into a shared $q$-dimensional space using respective MLPs, $MLP_c(\cdot)$ and $MLP_p(\cdot)$, and concatenated as

$$MLP_p(\boldsymbol{P})||MLP_c(\boldsymbol{x}) \in \mathbb{R}^{(l_p+1)\times q} \tag{5}$$

Simlarly to LORAX, the transformer $f$ processes this sequence and produces a joint representation `<cls>`

$$f(MLP_p(\boldsymbol{P})||MLP_c(\boldsymbol{x})) = MLP(\texttt{<cls>}) = y_{int} \tag{6}$$

where $y_{int}$ is an intermediate odorant-receptor affinity prediction. The $\texttt{<cls>}$ token is then combined with molecular and protein representations via an ensemble of XGBoost models:

$$y_{fin} = c_1 * h_1(\texttt{<cls>}) + c_2 * h_2(\boldsymbol{p}||\boldsymbol{x}) + c_3 * h_3(\texttt{<cls>}||\boldsymbol{p}||\boldsymbol{x}) \tag{7}$$

where $y_{fin}$ is the final prediction of the model, $h_{1,2,3}$ are the XGBoost models, and the ensemble weights $c_i$ satisfy $\sum_i c_i = 1$. This model is also trained using nested 5-fold cross validation. It is trained in a two step fashion, where the best performing transformer model on the validation data is saved and used to generate the $\texttt{<cls>}$ token. Next, each XGBoost is tuned independently via hyperopt, using random search over 500 iterations. The hyperparameter space and ranges are detailed in Table 6. After hyperparameter selection, the XGBoost models are trained on the training set and evaluated on the validation set to generate ensemble weights $c_1$, $c_2$, and $c_3$. Finally, the weighted ensemble is used to predict affinities.

## C  GCN MODEL

In addition to the models highlighted in Appendix B, we built a graph convolutional neural network (GCN, (Kipf & Welling, 2017)) to predict receptor responses from odorants alone. This method is similar to the MO model, but uses a graph representation as an input feature, and uses a GCN to predict odorant responses given a receptor.

The GCN predicts affinity using only molecular features. For each receptor, we train the GCN to take, as input, a graph $\mathcal{G} = (V, E)$, where a node vector $\boldsymbol{x}_i \in V$ is a 30-dimensional vector generated using deepchem (Ramsundar et al., 2019), and the connectivity of the graph is described using an adjacency matrix $A$ where $A_{ij} = 1$ if $(e_i, e_j) \in E$ and $A_{ij} = 0$ if $(e_i, e_j) \notin E$.

For each dataset, for each protein, we randomly split the data into train, validation, and test graphs. For the M2OR (EC50) dataset, as for the MO model, we use a random stratified splitting strategy to ensure there are both positive and negative examples in all data splits and ignore proteins with less than 50 examples. We train 5 GCN's per receptor using a 5-fold cross validation strategy, and report the average performance of all GCNs, averaged over folds and over all proteins in the dataset in Table 5. These results confirm, as we note in the main text, that using molecular features alone are insufficient for affinity prediction, and the use of protein information is necessary to create a reliably predictive model.

Table 5: Summary of GCN results. We report average $R^2$ values for the Hallem and Carey datasets and Matthews correlation coefficient (MCC) for the M2OR (EC50) dataset, each $\pm$ standard deviation computed across 5-fold cross validation and across proteins within each dataset.

| Dataset | GCN performance |
|---|---|
| Hallem (Hallem & Carlson (2006)) | $-0.0538 \pm 0.255$ |
| Carey (Carey et al. (2010)) | $-0.0933 \pm 0.781$ |
| M2OR (EC50) (Lalis et al. (2024)) | $0.0032 \pm 0.051$ |

## D  METRIC SPACE DISTANCE CALCULATION DETAILS

Here we provide a brief overview of how the distance metrics in the paper were calculated. Consider two odorant representations (e.g., CATS and ECFP) that map an odorant to feature vectors $\boldsymbol{x}_i \in \mathbb{R}^n$ and $\boldsymbol{x}_j \in \mathbb{R}^m$. Embedding an entire dataset of $N$ odorants with each representation yields matrices $\boldsymbol{X}_i \in \mathbb{R}^{N \times n}$ and $\boldsymbol{X}_j \in \mathbb{R}^{N \times m}$. To project each representation into the same space, we use PCA to reduce the dimensionality of both matrices to $k = 42$, resulting in $\boldsymbol{X}_i, \boldsymbol{X}_j \in \mathbb{R}^{N \times k}$.[9] We then optimize the following

---

[9]We use 42 here because the largest representation vector, 'ScaffoldKeyCalculator', is 42 length.

$$Q^* = \underset{Q \in \mathcal{O}}{\operatorname{argmin}} ||X_i - X_j Q|| \tag{8}$$

where $\mathcal{O}$ is the set of $k \times k$ orthogonal matrices. This optimization problem can be solved by doing a singular value decomposition on $X_i^\top X_j$ where $Q^* = UV^\top$. We calculate distance between the two spaces with

$$d(X_i, X_j) = \arccos\left(\frac{1}{k}\sum_{i=1}^{k}\sigma_i\right) \tag{9}$$

where $\sigma$ are the singular values of $X_i^\top X_j$. Optimizing equation 8 and calculating distance from equation 9 gives rise to the orthogonal Procrustes distance metric mentioned in the main text. If we instead whiten each representation as

$$X_i^\theta = X_i(X_i^\top X_i)^{-1/2}, \quad X_j^\theta = X_j(X_j^\top X_j)^{-1/2} \tag{10}$$

and solve

$$Q^* = \underset{Q \in \mathcal{O}}{\operatorname{argmin}} ||X_i^\theta - X_j^\theta Q|| \tag{11}$$

the resulting distances (again calculated via equation 9) correspond to the CCA distance metrics reported in the main text. These metrics are calculated using the package 'netrep' from Williams et al. (2022).

## E    NEURAL REPRESENTATION GENERATION

In this section we outline how the 'neural_representation' feature space is created for Figure 5 and Figure 7. The Carey dataset can be represented as a matrix $X \in \mathbb{R}^{N \times m}$, where $N$ is the number of odorants and $m$ is the number of receptors. Each odorant is described by its response profile across all receptors, so the feature vector for odorant $i$ is the $i$-th row of $X$, denoted $x_i \in \mathbb{R}^m$. This makes the entire matrix $X$ a representation of all odorants defined by their activation patterns across every receptor. This feature matrix $X$ is what we call 'neural_representation', and it can be directly compared to any other feature matrix $Y \in \mathbb{R}^{N \times k}$ (e.g. CATS or ECFP) to assess alignment. We use CCA and Procrustes alignment on the 'neural_representation' and all representations of odorants used in the paper to determine how aligned the featrure spaces are.

## F    LORAX MODEL DETAILS

In this section, we describe model details of our LORAX model. LORAX uses the Huggingface and PEFT libraries to adapt both the chemical and protein foundation models. All results shown in this work (unless otherwise specified) only adapt the chemical foundation model with LoRA. The chemical foundation model adapter was run with $r = 8$ rank adapter matrices on the query, key and value matrices with $\alpha = 8$. We do a hyperparameter sweep over rank of the adapter matrices and show results in Table S3. We do not train the bias parameters. We have a dropout rate of 0.1 in the adapter matrices during training. We do not use rslora (Kalajdzievski, 2023) as we are interested in starting from the original chemical foundation model representation to see how the representation changes over training. The cross attention block of the model is composed of two multi-headed attention layers. Both of these layers use 8 heads by default. The SMILES multi-headed attention layer takes the last hidden state of the LoRA adapted chemical foundation model ($\text{smi}_{lora}$) as the query, and the last hidden state of the protein foundation model ($\text{prot}_{ori}$) as the key and value matrices. The protein multi-headed attention layer does the opposite taking $\text{prot}_{ori}$ as the query matrix and $\text{smi}_{lora}$ as the key and value matrices. In both of these multi-headed attention layers there is a dropout rate of 0.1 during training. After the cross attention block, there is a residual connection where the output of the SMILES (protein) multi-headed attention layer is

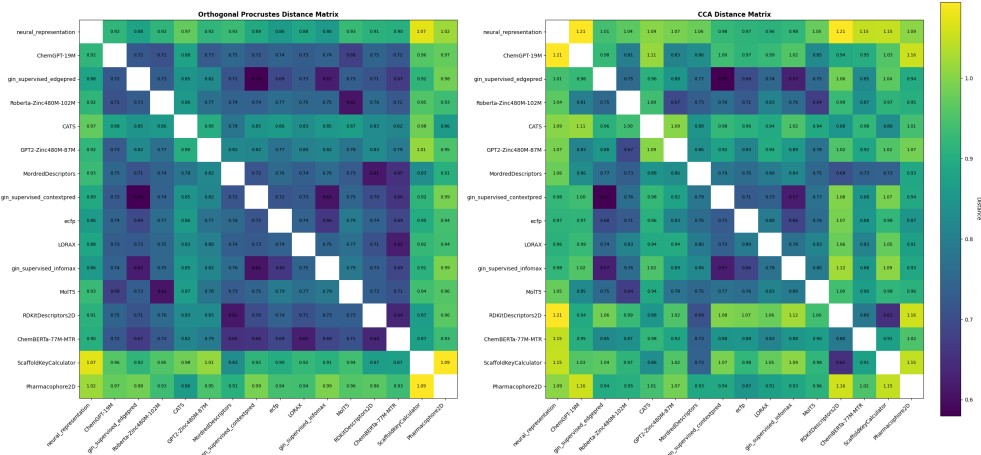

Figure 7: Full distance matrices using Procrustes (left) and CCA (right) distance metrics corresponding to the distance matrices shown in Figure 5.

added to $\text{smi}_{lora}$ ($\text{prot}_{ori}$). These representations are then passed through a layer norm. Both the protein and odorant representations are then mean pooled across their sequence dimensions and concatenated. This concatenation is the <cls> token referenced in the main text. The <cls> token is then put through an MLP with default dimension 512 to predict affinity between the odorant and receptor. During training, the model is run with a batch size of 12 for 50 epochs for the Carey, Hallem, and M2OR (EC50) datasets and batch size 21 for 15 epochs for the M2OR dataset. By default the training is conducted at a learning rate of 1e-5 unless otherwise specified. We use Adam as our optimizer. A block diagram for the LORAX model prior to the XGBoost ensemble is in Figure 8.

Throughout training, the loss on the validation set is monitored, and the weights of multi-modal transformer model with the lowest validation loss are saved. Then, <cls> tokens are generated for all data using that multi-modal transformer model. We then train 3 separate XGBoost models. One model takes, as input, the <cls> token; one takes, as input, the original, non LoRA-optimized, chemical foundation model concatenated to the protein foundation model representation; and the last model takes, as input, the <cls> token concatenated with the non LoRA-optimized chemical foundation model representation and the protein foundation model representation. Each XGBoost model has a set of hyperparameters that we optimize on the validation set using hyperopt. The hyperparameters, as well as the range of their values, are highlighted in Table 6. The parameters are optimized using a random search over a default of 500 iterations. After optimizing hyperparameters for all three XGBoost models, they are weighted according to their accuracy on the validation dataset. Finally, all XGBoost models are used to predict the test set values using the confidence weights. Code for this model is available here.

## G  ALL DATASETS AND METRICS STUDIED

This section presents all metrics and datasets not fully reported in Section 3. Tables 7 and 8 provide affinity prediction results and additional ProSmith metrics, respectively, for the Hallem dataset. Table 9 reports further metrics for the ProSmith model trained on the Carey dataset. Tables 10 and 11 show affinity prediction performance and other relevant metrics for the ProSmith model on the M2OR (EC50) dataset. Finally, Table 12 highlights general metrics for each dataset used in the paper.

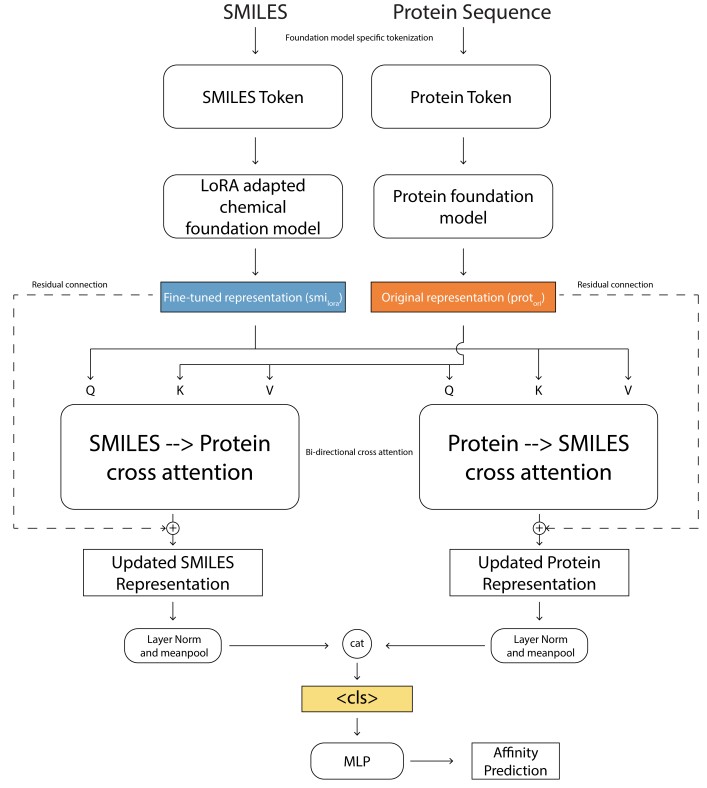

Figure 8: Block diagram for the LORAX model prior to the XGBoost ensemble.

Table 6: Hyperparameters and their search ranges for XGBoost optimization.

| Hyperparameter | Search Range |
| --- | --- |
| learning_rate | $0.01 - 0.5$ |
| max_depth | 6, 7, 8, 9, 10, 11, 12, 13, 14 |
| reg_lambda | $0 - 5$ |
| reg_alpha | $0 - 5$ |
| max_delta_step | $0 - 5$ |
| min_child_weight | $0.1 - 15$ |
| num_rounds | $30 - 1000$ |
| weight | $0.01 - 0.99$ |

Table 7: Affinity prediction performance across odorant representations for the Hallem dataset. Reported as mean $R^2 \pm$ standard deviation across 5 fold cross validation. Best performing representations are highlighted in bold.

| Odorant representation | MO | MP | ProSmith |
|---|---|---|---|
| CATS | -0.129 ± 1.819 | 0.367 ± 0.033 | 0.561 ± 0.032 |
| MordredDescriptors | 0.137 ± 0.341 | 0.382 ± 0.042 | 0.579 ± 0.036 |
| Pharmacophore2D | 0.192 ± 0.247 | 0.389 ± 0.037 | 0.530 ± 0.059 |
| RDKitDescriptors2D | 0.044 ± 0.201 | 0.302 ± 0.043 | **0.601 ± 0.057** |
| ScaffoldKeyCalculator | -0.014 ± 0.219 | 0.272 ± 0.040 | 0.501 ± 0.044 |
| ECFP | 0.182 ± 0.282 | 0.389 ± 0.036 | 0.558 ± 0.036 |
| Roberta-Zinc480M-102M | 0.183 ± 0.252 | **0.398 ± 0.039** | 0.564 ± 0.041 |
| GPT2-Zinc480M-87M | 0.095 ± 0.205 | 0.367 ± 0.041 | 0.521 ± 0.067 |
| ChemGPT-19M | 0.118 ± 0.240 | 0.359 ± 0.035 | 0.559 ± 0.051 |
| MolT5 | 0.156 ± 0.249 | 0.389 ± 0.035 | 0.555 ± 0.046 |
| ChemBERTa-77M-MTR | 0.198 ± 0.349 | 0.373 ± 0.039 | 0.574 ± 0.040 |
| gin_supervised_infomax | 0.198 ± 0.287 | 0.396 ± 0.041 | 0.530 ± 0.044 |
| gin_supervised_edgepred | 0.177 ± 0.252 | 0.390 ± 0.040 | 0.550 ± 0.031 |
| gin_supervised_contextpred | **0.203 ± 0.411** | 0.396 ± 0.043 | 0.547 ± 0.061 |
| random | -0.069 ± 0.119 | 0.382 ± 0.039 | 0.439 ± 0.073 |

Table 8: Other metrics for the ProSmith model trained on the Hallem dataset. Error indicates standard deviation across 5 fold cross validation. Best performing representations are highlighted in bold. MSE: mean squared error, rm2: modified $R^2$ defined as $r_m^2 = r^2(1 - \sqrt{r^2 - r_0^2})$ where $r^2$ and $r_0^2$ are correlation coeficients with and without intercept respectfully (Roy et al., 2013), CI: concordance index.

| Odorant representation | MSE | rm2 | CI |
|---|---|---|---|
| CATS | 0.462 ± 0.042 | 0.548 ± 0.034 | 0.818 ± 0.006 |
| MordredDescriptors | 0.444 ± 0.052 | 0.571 ± 0.043 | 0.824 ± 0.009 |
| Pharmacophore2D | 0.494 ± 0.068 | 0.526 ± 0.052 | 0.806 ± 0.007 |
| RDKitDescriptors2D | **0.421 ± 0.075** | **0.587 ± 0.061** | **0.830 ± 0.007** |
| ScaffoldKeyCalculator | 0.524 ± 0.050 | 0.491 ± 0.044 | 0.813 ± 0.011 |
| ECFP | 0.466 ± 0.049 | 0.551 ± 0.034 | 0.821 ± 0.004 |
| Roberta-Zinc480M-102M | 0.459 ± 0.053 | 0.554 ± 0.046 | 0.824 ± 0.005 |
| GPT2-Zinc480M-87M | 0.505 ± 0.081 | 0.511 ± 0.062 | 0.810 ± 0.011 |
| ChemGPT-19M | 0.465 ± 0.068 | 0.551 ± 0.058 | 0.821 ± 0.008 |
| MolT5 | 0.470 ± 0.066 | 0.546 ± 0.052 | 0.818 ± 0.010 |
| ChemBERTa-77M-MTR | 0.449 ± 0.055 | 0.563 ± 0.045 | 0.821 ± 0.004 |
| gin_supervised_infomax | 0.494 ± 0.052 | 0.522 ± 0.045 | 0.820 ± 0.007 |
| gin_supervised_edgepred | 0.474 ± 0.045 | 0.541 ± 0.031 | 0.826 ± 0.003 |
| gin_supervised_contextpred | 0.478 ± 0.074 | 0.533 ± 0.060 | 0.819 ± 0.007 |
| random | 0.590 ± 0.086 | 0.434 ± 0.071 | 0.793 ± 0.012 |

Table 9: Other metrics for the ProSmith model trained on the Carey dataset. Error indicates standard deviation across 5 fold cross validation. Best performing representations are highlighted in bold. MSE: mean squared error, rm2: modified $R^2$ defined as $r_m^2 = r^2(1 - \sqrt{r^2 - r_0^2})$ where $r^2$ and $r_0^2$ are correlation coeficients with and without intercept respectfully (Roy et al., 2013), CI: concordance index.

| Odorant representation | MSE | rm2 | CI |
|---|---|---|---|
| CATS | $0.378 \pm 0.036$ | $0.631 \pm 0.016$ | $0.815 \pm 0.008$ |
| MordredDescriptors | $0.293 \pm 0.037$ | $0.717 \pm 0.027$ | $\mathbf{0.822 \pm 0.005}$ |
| Pharmacophore2D | $0.471 \pm 0.038$ | $0.543 \pm 0.011$ | $0.801 \pm 0.008$ |
| RDKitDescriptors2D | $\mathbf{0.292 \pm 0.039}$ | $\mathbf{0.718 \pm 0.028}$ | $0.821 \pm 0.009$ |
| ScaffoldKeyCalculator | $0.422 \pm 0.031$ | $0.590 \pm 0.023$ | $0.796 \pm 0.012$ |
| ECFP | $0.351 \pm 0.046$ | $0.658 \pm 0.030$ | $0.805 \pm 0.010$ |
| Roberta-Zinc480M-102M | $0.341 \pm 0.037$ | $0.666 \pm 0.031$ | $0.817 \pm 0.011$ |
| GPT2-Zinc480M-87M | $0.347 \pm 0.045$ | $0.661 \pm 0.030$ | $0.813 \pm 0.011$ |
| ChemGPT-19M | $0.328 \pm 0.034$ | $0.680 \pm 0.024$ | $0.816 \pm 0.014$ |
| MolT5 | $0.338 \pm 0.038$ | $0.669 \pm 0.021$ | $0.812 \pm 0.005$ |
| ChemBERTa-77M-MTR | $0.309 \pm 0.032$ | $0.700 \pm 0.020$ | $0.822 \pm 0.010$ |
| gin_supervised_infomax | $0.343 \pm 0.034$ | $0.667 \pm 0.018$ | $0.812 \pm 0.013$ |
| gin_supervised_edgepred | $0.326 \pm 0.035$ | $0.681 \pm 0.028$ | $0.820 \pm 0.011$ |
| gin_supervised_contextpred | $0.315 \pm 0.047$ | $0.696 \pm 0.035$ | $\mathbf{0.822 \pm 0.008}$ |
| random | $0.431 \pm 0.047$ | $0.582 \pm 0.038$ | $0.796 \pm 0.011$ |

Table 10: Affinity prediction performance across odorant representations for the M2OR (EC50) dataset. As stated in the main text, this dataset it a subset of of the M2OR dataset using on EC50 labeled data. Reported as mean Matthew's correlation coefficient (MCC) $\pm$ standard deviation. Best performing representations are highlighted in bold.

| Odorant representation | MO | MP | ProSmith |
|---|---|---|---|
| CATS | $0.090 \pm 0.262$ | $0.562 \pm 0.030$ | $0.674 \pm 0.029$ |
| MordredDescriptors | $0.113 \pm 0.291$ | $0.568 \pm 0.027$ | $\mathbf{0.709 \pm 0.018}$ |
| Pharmacophore2D | $0.097 \pm 0.270$ | $0.570 \pm 0.030$ | $0.656 \pm 0.032$ |
| RDKitDescriptors2D | $0.066 \pm 0.241$ | $0.515 \pm 0.038$ | $0.694 \pm 0.024$ |
| ScaffoldKeyCalculator | $0.053 \pm 0.198$ | $0.583 \pm 0.025$ | $0.646 \pm 0.032$ |
| ECFP | $0.096 \pm 0.293$ | $0.581 \pm 0.030$ | $0.663 \pm 0.016$ |
| Roberta-Zinc480M-102M | $0.131 \pm 0.310$ | $0.568 \pm 0.029$ | $0.697 \pm 0.027$ |
| GPT2-Zinc480M-87M | $0.109 \pm 0.286$ | $0.563 \pm 0.026$ | $0.678 \pm 0.023$ |
| ChemGPT-19M | $0.103 \pm 0.295$ | $0.574 \pm 0.035$ | $0.683 \pm 0.017$ |
| MolT5 | $0.113 \pm 0.294$ | $0.589 \pm 0.035$ | $0.685 \pm 0.030$ |
| ChemBERTa-77M-MTR | $\mathbf{0.172 \pm 0.351}$ | $0.591 \pm 0.030$ | $0.696 \pm 0.033$ |
| gin_supervised_infomax | $0.109 \pm 0.300$ | $\mathbf{0.595 \pm 0.030}$ | $0.684 \pm 0.030$ |
| gin_supervised_edgepred | $0.118 \pm 0.312$ | $0.593 \pm 0.021$ | $0.692 \pm 0.040$ |
| gin_supervised_contextpred | $0.103 \pm 0.266$ | $0.589 \pm 0.031$ | $0.695 \pm 0.026$ |
| random | $0.004 \pm 0.120$ | $0.550 \pm 0.029$ | $0.640 \pm 0.027$ |

Table 11: Other metrics for the ProSmith model trained on the M2OR (EC50) dataset. Error indicates standard deviation across 5 fold cross validation. Best performing representations are highlighted in bold. AUROC: area under the receiver operating characteristic curve.

| Odorant representation | Accuracy | AUROC |
|---|---|---|
| CATS | $0.877 \pm 0.013$ | $0.917 \pm 0.006$ |
| MordredDescriptors | $\mathbf{0.892 \pm 0.006}$ | $\mathbf{0.933 \pm 0.007}$ |
| Pharmacophore2D | $0.872 \pm 0.012$ | $0.913 \pm 0.007$ |
| RDKitDescriptors2D | $0.886 \pm 0.008$ | $0.931 \pm 0.007$ |
| ScaffoldKeyCalculator | $0.870 \pm 0.012$ | $0.910 \pm 0.005$ |
| ECFP | $0.873 \pm 0.005$ | $0.923 \pm 0.009$ |
| Roberta-Zinc480M-102M | $0.889 \pm 0.010$ | $0.927 \pm 0.005$ |
| GPT2-Zinc480M-87M | $0.882 \pm 0.008$ | $0.923 \pm 0.005$ |
| ChemGPT-19M | $0.883 \pm 0.006$ | $0.923 \pm 0.006$ |
| MolT5 | $0.884 \pm 0.010$ | $0.925 \pm 0.007$ |
| ChemBERTa-77M-MTR | $0.888 \pm 0.012$ | $0.927 \pm 0.003$ |
| gin_supervised_infomax | $0.884 \pm 0.010$ | $0.925 \pm 0.006$ |
| gin_supervised_edgepred | $0.887 \pm 0.013$ | $0.928 \pm 0.008$ |
| gin_supervised_contextpred | $0.887 \pm 0.010$ | $0.925 \pm 0.004$ |
| random | $0.870 \pm 0.010$ | $0.902 \pm 0.007$ |

Table 12: Summary table for all datasets used in this work. $n_R$ refers to the number of unique receptors in the dataset, $n_O$ refers to the number unique of odorants, and pairs refers to the number of recptor-odorant pairs in each dataset.

| Dataset | $n_R$ | $n_O$ | Pairs | Response |
|---|---|---|---|---|
| Hallem (Hallem & Carlson (2006)) | 24 | 110 | 2640 | spikes/s |
| Carey (Carey et al. (2010)) | 50 | 110 | 5500 | spikes/s |
| M2OR (EC50) (Lalis et al. (2024)) | 508 | 474 | 5835 | binary |
| M2OR (Lalis et al. (2024)) | 1237 | 596 | 46563 | binary |

