# Supplement for the paper

# Low Rank Adaptation of Chemical Foundation Models Generate Effective Odorant Representations

November 2025

Table 1: Distribution of responsive vs. non-responsive classes in the M2OR dataset.

| M2OR Dataset | Responsive | Non-Responsive | % Responsive |
|---|---|---|---|
| EC50 (highest quality) | 1,153 | 4,062 | 28.4% |
| Secondary screening (middle) | 283 | 7,875 | 3.6% |
| Primary screening (lowest) | 1,464 | 31,726 | 4.6% |
| All Data | 2,900 | 43,663 | 6.6% |

Table 2: Predictive performance using a protein-only model. For each test dataset, the model uses only protein features for prediction. Values show mean $\pm$ standard deviation across five cross-validation folds. $R^2$: coefficient of determination, MCC: Matthews correlation coefficient.

| Dataset | Protein-Only Prediction |
|---|---|
| Hallem Dataset ($R^2$) | $-3.56 \pm 24.1$ |
| Carey Dataset ($R^2$) | $-0.336 \pm 1.60$ |
| M2OR (EC50) (MCC) | $0.098 \pm 0.307$ |

For the M2OR (EC50) dataset, odorants with fewer than 23 proteins were excluded from training, due to limited data.

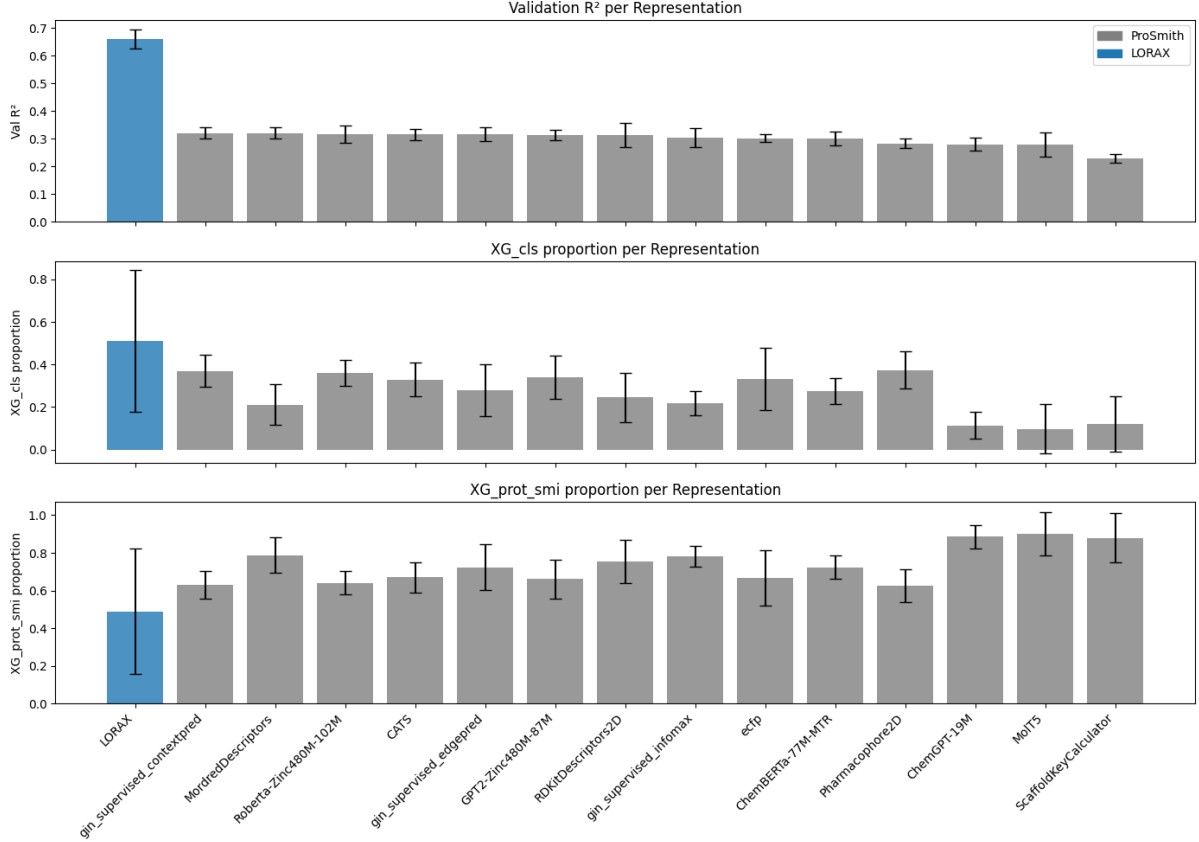

Figure 1: Comparison of LORAX and ProSmith representations after training two XGBoost models on the Carey dataset: one using only the `<cls>` token, and one using the original chemical and protein foundation model representations. (Top) Mean validation $R^2$ scores for ProSmith and LORAX prior to applying XG-Boost. (Middle) The mean weight assigned to the XGBoost model utilizing only the `<cls>` token. (Bottom) The mean weight assigned to the XGBoost model utilizing only the original chemical and protein foundation model representations. All error bars indicate standard deviation across 5-fold cross validation of the data.

Table 3: Hyperparameter sweep of rank ($r$) of LoRA adapter for the LORAX model. The model was trained on the Carey dataset. Reported values are test set $R^2$ across folds without the XGBoost ensemble. Bolded value indicates highest performance. All models were trained with query, key, and value weight matrices being adapted.

| Rank | rand_split_1 | rand_split_2 | rand_split_3 | rand_split_4 | rand_split_5 | avg | std |
|------|------|------|------|------|------|------|------|
| $r = 2$ | 0.6456 | 0.5921 | 0.5045 | 0.6029 | 0.6134 | **0.5917** | 0.0471 |
| $r = 4$ | 0.6179 | 0.5802 | 0.5009 | 0.5884 | 0.5621 | 0.5699 | 0.0389 |
| $r = 8$ | 0.6267 | 0.6115 | 0.4986 | 0.5892 | 0.6082 | 0.5868 | 0.0457 |
| $r = 16$ | 0.6122 | 0.5965 | 0.5183 | 0.5506 | 0.6212 | 0.5798 | 0.0392 |

Table 4: Hyperparameter sweep of rank LoRA adapted modules for the LORAX model. The model was trained on the Carey dataset. Reported values are test set $R^2$ across folds without the XGBoost ensemble. Bolded value indicates highest performance. $W_q$, $W_k$, and $W_v$, indicate query, key, and value weight matrices respectively. All models were trained with rank 8 adapter matrices.

| Adapted Modules | rand_split_1 | rand_split_2 | rand_split_3 | rand_split_4 | rand_split_5 | avg | std |
|---|---|---|---|---|---|---|---|
| $W_q$ | 0.6426 | 0.6148 | 0.5505 | 0.5727 | 0.6018 | **0.5964** | 0.0322 |
| $W_q, W_k$ | 0.6437 | 0.6113 | 0.5162 | 0.5674 | 0.6085 | 0.5894 | 0.0439 |
| $W_q, W_v$ | 0.6312 | 0.5993 | 0.5446 | 0.5700 | 0.5908 | 0.5872 | 0.0290 |
| $W_q, W_k, W_v$ | 0.6267 | 0.6115 | 0.4986 | 0.5892 | 0.6082 | 0.5868 | 0.0457 |

Table 5: Pairwise post hoc comparisons between models for unseen odorants (as presented in Table 2, main text). Results are based on Conover's signed-rank test following a significant Friedman test; all $p$-values were corrected for multiple comparisons using the Benjamini-Hochberg false discovery rate (FDR) procedure.

| | | T-Stat | df | $W_i$ | $W_j$ | $r_{rb}$ | p | $p_{FDR}$ |
|---|---|---|---|---|---|---|---|---|
| LORAX | ProSmith | 2.353 | 8 | 14.000 | 8.000 | 1.000 | .046 | .069 |
| | Naive | 2.353 | 8 | 14.000 | 8.000 | 0.467 | .046 | .069 |
| ProSmith | Naive | 0.000 | 8 | 8.000 | 8.000 | 0.067 | 1.000 | 1.000 |

Table 6: Pairwise post hoc comparisons between models for unseen receptors (as presented in Table 2, main text). Results are based on Conover's signed-rank test following a significant Friedman test; all $p$-values were corrected for multiple comparisons using the Benjamini-Hochberg false discovery rate (FDR) procedure.

| | | T-Stat | df | $W_i$ | $W_j$ | $r_{rb}$ | p | $p_{FDR}$ |
|---|---|---|---|---|---|---|---|---|
| LORAX | ProSmith | 0.000 | 8 | 12.00 | 12.000 | $-0.067$ | 1.000 | 1.000 |
| | Naive | 2.353 | 8 | 12.00 | 6.000 | 1.000 | .046 | .069 |
| ProSmith | Naive | 2.353 | 8 | 12.00 | 6.000 | 0.867 | .046 | .069 |

Table 7: Pairwise post hoc comparisons between Matthews correlation coefficient (MCC) values for LORAX and ProSmith models on M2OR in Table 3 of the main text. Results are based on Conover's signed-rank test following a significant Friedman test; all p-values were corrected for multiple comparisons using the Benjamini-Hochberg false discovery rate (FDR) procedure.

| | | T-Stat | df | $W_i$ | $W_j$ | $r_{rb}$ | p | $p_{FDR}$ |
|---|---|---|---|---|---|---|---|---|
| LORAX | ProSmith | 2.121 | 8 | 14.00 | 11.000 | 0.733 | .067 | 0.067 |
| | MolOR (Weighted) | 6.364 | 8 | 14.00 | 5.000 | 1.000 | $< .001$ | $< 0.003$ |
| ProSmith | MolOR (Weighted) | 4.243 | 8 | 11.00 | 5.000 | 1.000 | .003 | 0.0045 |

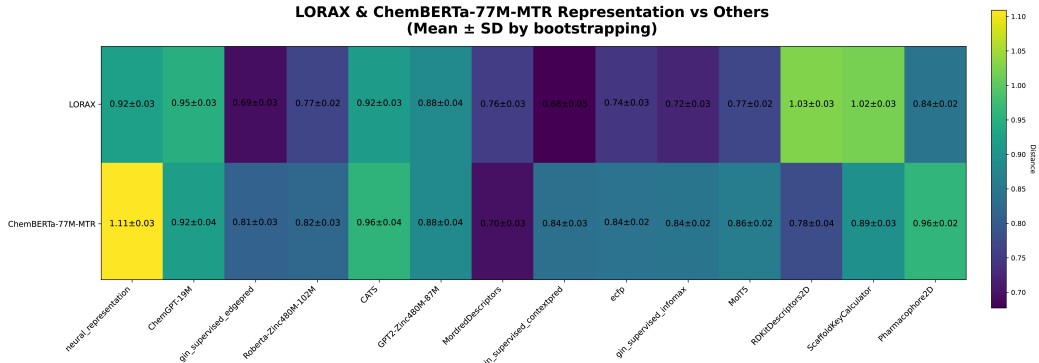

Figure 2: CCA distance matrix for LORAX and ChemBERTa-77M-MTR representations on the Carey dataset, estimated via 1,000 bootstraped samples. Each cell reports the mean ± standard deviation of distances computed over the 1,000 smaples. For each bootstrap iteration, 110 odorants were randomly selected with replacement, and CCA distances were computed between the corresponding feature spaces. Bootstrapped CCA distances between LORAX and ChemBERTa-77M-MTR representations and all other models were compared using Welch's t-tests, with multiple comparisons controlled via false discovery rate (FDR) correction. All pairwise differences were statistically significant (p < 0.0001), except for those involving GPT2-Zinc480M-87M.

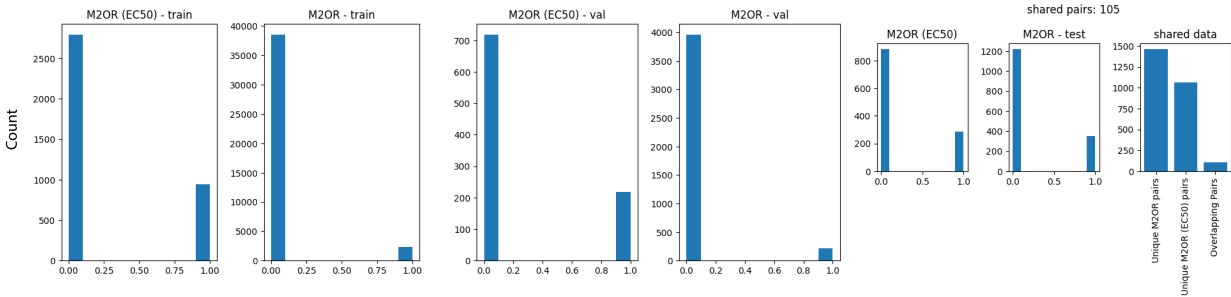

Figure 3: Distributions of train, validation, and test splits for split 1 in both the M2OR and M2OR (EC50) datasets. Although the test sets share similar overall distributions, substantial differences are observed in the train and validation sets. Furthermore, the test set composition varies considerably between the two datasets.

Table 8: Comparison of various LORAX models trained on the M2OR dataset. LORAX (C) and LORAX (P + C) are taken from the main text Table 3. LORAX (U) uses Uni-Mol2-84M (Ji et al., 2024) as the chemical foundation model instead of ChemBERTa-77M-MTR. Uncertainties for all models are standard deviation across 5 fold cross validation. Bold indicates highest value in each column. n/a entries indicate that the authors did not report that metric. AUROC: area under the receiver operating characteristic curve, AveP: average precision, MCC: Matthews correlation coefficient.

| | AUROC | AveP | Precision | Recall | F-score | MCC |
|---|---|---|---|---|---|---|
| LORAX (U) | $80.36 \pm 1.02$ | $0.775 \pm 0.03$ | $\mathbf{0.732 \pm 0.05}$ | $0.679 \pm 0.02$ | $0.703 \pm 0.02$ | $0.624 \pm 0.03$ |
| LORAX (P + C) | $82.43 \pm 0.86$ | $0.776 \pm 0.03$ | $0.729 \pm 0.06$ | $0.727 \pm 0.03$ | $0.727 \pm 0.02$ | $0.650 \pm 0.03$ |
| LORAX (C) | $\mathbf{83.24 \pm 1.34}$ | $\mathbf{0.778 \pm 0.03}$ | $0.710 \pm 0.05$ | $\mathbf{0.754 \pm 0.02}$ | $\mathbf{0.730 \pm 0.03}$ | $\mathbf{0.651 \pm 0.04}$ |

Table 9: Performance of LORAX and a Naïve Baseline on Hallem and M2OR (EC50) Datasets for Unseen Odorants and Receptors. Metrics for each split are reported in $R^2$ for the Hallem dataset and Matthews Correlation Coefficient (MCC) for the M2OR (EC50) dataset. For the Naïve baselines, predictions for the Hallem dataset were obtained by calculating the mean of the training set, and using that value as a guess for every test example. For the M2OR (EC50) dataset, Naïve predictions were generated by randomly assigning either 0 or 1 to each test example with equal probability.

| Dataset | Scenario | Method | rand_split_1 | rand_split_2 | rand_split_3 | rand_split_4 | rand_split_5 | avg | std |
|---|---|---|---|---|---|---|---|---|---|
| Hallem | Unseen odorants | LORAX | -0.599 | -1.138 | 0.250 | 0.292 | 0.421 | **-0.155** | 0.610 |
| | | Naïve | -0.499 | -0.394 | -0.003 | -0.102 | -0.076 | -0.215 | 0.195 |
| Hallem | Unseen receptors | LORAX | 0.018 | -0.021 | 0.087 | 0.151 | -2.358 | -0.425 | 0.968 |
| | | Naïve | -0.049 | -0.812 | -0.001 | -0.087 | -0.052 | **-0.200** | 0.307 |
| M2OR (EC50) | Unseen odorants | LORAX | 0.634 | 0.736 | 0.522 | 0.622 | 0.310 | **0.565** | 0.144 |
| | | Naïve | 0.078 | -0.030 | -0.041 | 0.078 | -0.033 | 0.010 | 0.055 |
| M2OR (EC50) | Unseen receptors | LORAX | 0.351 | 0.295 | 0.389 | 0.149 | 0.244 | **0.286** | 0.0841 |
| | | Naïve | -0.043 | -0.037 | -0.005 | -0.010 | 0.023 | -0.014 | 0.024 |