# OpenReview forum: "Low rank adaptation of chemical foundation models generate effective odorant representations"
_ICLR.cc/2026/Conference — ICLR 2026 Poster_

### Official Review · Reviewer_P2Me · 2025-10-25

**Soundness:** 2
**Presentation:** 1
**Contribution:** 1
**Rating:** 2
**Confidence:** 3

**Summary:**

The paper presents a benchmark of several chemical foundation models in the scope of the notoriously difficult problem of modelling olfaction. The authors showed that the zero-shot application of chemical foundation models does not increase the performance in molecule-receptor response prediction tasks compared to using standard physicochemical features. Thus, to mitigate this, the authors suggest using LoRA to fine-tune chemical foundation models (e.g. ChemBERTa-77M-MTR) to model interactions at the beginning of the olfactory system. The authors demonstrate that combining LoRA with ProSmith model (introduced in Kroll et al., 2024) leads to 1.5\% increase in the performance on the random splits of the challenging M2OR dataset compared to ProSmith without fine-tuning.

**Strengths:**

The paper made an interesting observation, that embedding of chemical foundation models is similar to the traditional physicochemical descriptors. In addition, the proposed model, LORAX, achieves a marginal increase in the SOTA preformance on M2OR dataset compared to ProSmith model which does not use LoRA fine-tuning.

**Weaknesses:**

The proposed algorithm, LORAX, is rather incremental and only adds a well-established LoRA fine-tuning to ChemBERTa in the previously-published ProSmith model. The most interesting result of the paper is the observation that embedding spaces of the tested chemical foundation models are aligned with each other and with standard physicochemical descriptors. This is a surprising result, given that chemical foundation models are widely-studied and have been shown to outperform physicochemical descriptors in molecular properties prediction. However, the authors do not provide any explanation of such an unexpected phenomenon, making the result much less impactful. Additionally, the authors do not provide the code, so I am unable to reproduce the results, nor check the data splitting strategies.

**Questions:**

Some of the experiments are not clear. For instance, how are Table 3 and Table 9 related? Both tables discuss the performance on M2OR dataset. Assuming that the authors evaluate the models in Table 3 on "ec50" pairs due to the label noise (following Hladi\v{s} et al.), I would expect that the values in the tables are comparable. However, "MorderdDescriptors" achieves MCC of 0.709 in Table 9, outperforming the proposed LORAX approach in Table 3.

In Figure 5 and Figure 7, the authors compare LORAX,  ChemBERTa, and other odorant representations to the "neural representation" derived from Carey dataset. However, in my understanding, this comparison is not fair, since LORAX is trained on the neuronal activity data and other models are zero-shot. In addition, despite being trained, LORAX does not appear to be significantly more aligned with neuronal activity than, zero-shot "ecfp" or "gin\_supervised\_infomax" representations (first columns in Figure 7).

Since the authors do not provide the code during the review process, I am unable to check data splitting, which is crucial in the field of protein-molecule interaction prediction. This said, how are the unseen receptor/molecule splits constructed in Table 2? The similarity of the protein sequences/molecular structure between the train and test sets can largely mislead the performance evaluation (see e.g. Zhang et al. Rethinking the generalization of drug target affinity prediction algorithms via similarity aware evaluation, ICLR 2025). In addition, the generalization experiment is conducted on the Carey dataset. What is the performance on other two datasets? In particular, how this compares to the generalization performance presented in Hladi\v{s} et al.?

Considering the style of writing, I find the paper imprecise and unpolished, especially in the use of terms such as "binding", "affinity" and "activation". Data in Hallem and Carey datasets are average spikes per second for insect electrophysiology experiments, and data in M2OR are mostly in vitro functional assays in heterologous systems. Neither of these experiments directly measure "binding" as binding of the molecule to the protein is only a necessary but not a sufficient condition to observe the experimental response. In the case of M2OR dataset, the term "affinity" is also imprecise, since the authors consider activation prediction, which is different from predicting a measure of affinity, such as EC50.

Section 4.1. is difficult to follow. At the beginning the authors write "We trained LORAX on the Carey dataset and found it performs almost identically to ProSmith" and then suddenly in Figure 4 LORAX performs better than ProSmith. Is the performance in Figure 4 only for the Transfomer part of ProSmith? If yes, then this is not clear from the text and the figure caption.

Lastly, I found several incorrect citations in the paper (Cong et al. 2022, Haddad et al. 2008a, Lalis et al. 2024, Sypetkowski et al. 2024). I believe that the errors come from some citation tool, but the authors should thoroughly check the citations as if passed unnoticed, this shows disrespect to the original authors.

---

> ### Author Response · Authors · 2025-11-18
> **Response to reviewer r6QY's initial review (1/2)**
>
> Thank you for carefully reading our work and noting our interesting finding of overlapping feature spaces. We break our rebuttal into two separate comments addressing weaknesses and questions respectively. Regarding the point raised about marginal improvement over the SOTA the reviewer made in their strengths, we refer the reviewer to our response to reviewer r6Qy (“W3: Limited Performance Gains of LORAX Over ProSmith”). Briefly, we note that ProSmith was not, prior to our work, considered SOTA for the M2OR dataset, but rather the Hladiš paper (benchmarked in Table 3) which we show significant improvements over. Given that ProSmith is not a well-established model in olfaction, we view benchmarking ProSmith as a result in and of itself, and believe focusing on modest performance gains over it does not fully capture the significance of our other contributions.
>
> # W1
> We respectfully contend that our contributions are both novel and significant for the ICLR community, and make our case below.
>
> ## LORAX is incremental, because LoRA is well-established
> We understand the reviewer’s perspective that \textit{in other domains} LoRA is well-established. To the best of our knowledge (consistent with reviewer r6Qy) we have not seen LoRA applied in olfaction, and thus contend our work is significant and novel. We will gladly cite existing demonstrations of this if they can be provided. But to the best of our knowledge, none exist.
>
> ## Our surprising, and thus interesting, result, lacking interpretation
> While prior models show strong performance on molecular property prediction, our surprising finding (as the reviewer noted), is that, for odorant-receptor prediction, their learned representations do not improve performance over physicochemical descriptors. This unexpected alignment suggests that, despite general performance gains, these models lack specificity for olfaction. We highlight an inconsistency here: the reviewer notes this is surprising and interesting, but also lists it as a weakness because of a lack of interpretation. We do our best to interpret these results (see below), but also wish to make the case that these surprising results, combined with our novel application of LoRA, we believe, make our work of interest to many in the ICLR community, and hope the reviewer will revisit their review in light of our response.
>
> Recognizing the shortcomings of existing chemical foundation models for olfaction, we developed LORAX. Unlike ProSmith, (where foundation embeddings are assumed sufficient for prediction), our method addresses the issue of embedding alignment. LORAX was \textit{directly} motivated by our observation that the feature spaces overlapped; had this not been the case, there would be no need to develop a new approach and we would simply have adopted ProSmith’s methodology. By interrogating why current representations fail to capture olfactory-relevant distinctions and proposing a tailored fine-tuning strategy, our work advances domain applicability of chemical foundation models. We believe this approach not only provides insight into when and how foundation model representations are effective, but also informs the community on strategies for adapting general-purpose models to olfaction.
>
> ## Interpretation
> We acknowledge an interpretation of this for olfaction would be nice, however we hesitate to make overly strong claims because of the complexity of the issue. Two points worth considering. 1) Our results do not invalidate previous work showing the improvement of these models of molecular property prediction: despite being partially aligned in this domain, they may still enable better molecular property prediction. 2) Our results speak more to the olfactory receptor prediction problem, which is not the same as chemical property prediction. It has been known for some time that chemical properties alone are insufficient to predict olfactory responses (either their odor profile, or how they interact with olfactory receptors [1, 2]). Our current interpretation of our result is consistent with this perspective. Olfactory response prediction is related, but not entirely identical, to the problem of molecular property prediction. We feel comfortable echoing this well-established perspective in our submission, if the reviewer feels this would provide further impact. However, we do not feel a lack of interpretation of these findings lessens the impact and application of our approach, which we feel will provide ML olfactory researchers new tools and improved insights about how chemical foundation models can be used to address the unique challenges presented by olfactory response prediction.
>
> ## Code
> To facilitate reproducibility, we updated our paper to include links to GitHub repositories containing code data, in the "Reproducibility Statement” section. The anonymous data download link in the readme expires on 11/22. If the reviewer misses this window, and needs another download link, please ask!

---

> > ### Author Response · Authors · 2025-11-18
> > **Response to reviewer r6QY's initial review (2/2)**
> >
> > # Q1
> > We apologize for this confusion. The tables in Appendix E are related to the datasets used in Section 3. These datasets are the Hallem dataset, Carey dataset, and the M2OR (EC50) dataset. The dataset in Table 3 is the entirety of the M2OR dataset, called “M2OR” in our paper. This is the same dataset used in Hladiš et al., following the exact splitting conventions they used as well. The “M2OR (EC50)” dataset in this work, is a subset of the entire M2OR dataset that only uses the highest quality data (i.e. the ec50 data) and is substantially smaller than the dataset as a whole. Would the reviewer recommend a way to make this more understandable, or is this explanation sufficient?
> >
> > # Q2
> > This is an interesting point and we are glad the reviewer brought it up. The reviewer is correct that LORAX is trained on the data we are comparing it to in the neural_representation column in Figure 5. We too believe this is unsurprising, as the model should, in theory, generate a representation that is more aligned.
> >
> > To the reviewer's second point, we were also surprised that some one-shot representations were more aligned with the neural representation than LORAX, although we have not yet fully characterized why this happens. We believe this may give insight into which models are ‘primed’ to have a better representation and thus should be targets of LoRA fine tuning. For example, we would be interested in fine-tuning “gin_supervised_infomax” to determine if LoRA fine-tuning would achieve even stronger results. We could endeavor to accomplish this within the review window, if the reviewer felt as though doing so would significantly increase the impact of our work, however, we note that there are significant technical limitations to doing this (see “W2: More recent models” in our response to Reviewer jDeV) that may make doing this in such a short time-window beyond our ability. We admit that there is some ambiguity here, and hope to further address this phenomenon in future studies. However, we do not believe the remaining ambiguity of this result overshadows the strong contributions noted above, namely the novel use of LoRA for chemical foundation models, the performance gains compared to ProSmith, the significant performance gains compared to Hladiš et al., and the novel insight that chemical foundation models struggle on olfaction prediction tasks, which to our knowledge, has not yet been reported.
> >
> > # Q3
> > We have added an anonymous link to our code and the datasets used in the most recent revision of the paper. The unseen receptor/odorant splits are done using CDHIT [3] and scaffold splitting respectively. For the unseen receptor splits, we first group similar receptors using CDHIT, then create splits ensuring similar receptors are not in both the training and test sets. For the unseen odorant we use deepchem’s ScaffoldSplitter, grouping molecules by their Bemis-Murcko scaffold before creating splits. This ensures that molecules in the train set do not have similar physical properties as molecules in the test set.
> >
> > # Q4
> > We thank the reviewer for this comment, and apologize for being imprecise in our language. Given the noted differences between the responses in the two datasets, we struggled to identify a common language that would not cognitively over-burden readers. We believe that “response activation” is sufficiently broad to capture both datasets, and also sufficiently distant from the specific terms "affinity" and “binding” to not misrepresent our work. Would the use of “activation” be acceptable? We are also open to additional suggestions.
> >
> > # Q5
> > We apologize for being unclear. The validation scores shown in Figure 4 are for the transformer part of both models prior to the XGBoost ensemble. We have changed the Figure caption and updated the text.
> >
> > # Q6
> > Thank you for noting this! We used an automated bibliography system from Zotero, and assumed all citations in the work were without error. We are extremely grateful that you pointed this out and have fixed all citations.
> >
> > # refs
> > [1] Lee, B. K. et al. Science 381, 999–1006 (2023). [2] Sell, C. S. Angewandte Chemie International Edition 45, 6254–6261 (2006). [3] Li, W. & Godzik, A. Bioinformatics 22, 1658–1659 (2006).

---

> > > ### Comment · Reviewer_P2Me · 2025-11-22
> > >
> > > First of all, I would like to thank the authors for their engagement in the review process. Below are responses to several questions, which I find the most crucial and which need clarification.
> > >
> > > ## W1 and W3 of reviewer r6Qy: Limited Performance Gains of LORAX Over ProSmith
> > > Although I appreciate the amount of work the authors invested in benchmarking the molecular representation, I would argue, that simply applying ProSmith to a new dataset can not be considered a significant contribution. Since 2023 when Hladi\v{s} et al. paper was published, several protein-molecule interaction models have been proposed that could be applied on M2OR dataset. Since this field is an active area of research, I believe that several new models would outperform Hladi\v{s} et al. On the other hand, it may be valuable to the community to test these models on the challenging M2OR dataset, but in that case, a comprehensive benchmark of all relevant recent models should be performed. In addition, as a minor point, I also agree with reviewer Cjx2, that the improvement over ProSmith may be questionable given the standard deviation and a statistical test should provide more information whether LORAX is indeed significantly increasing the SOTA performance.
> > >
> > > ## Q1
> > > I thank the authors for clarifying this point, and in my opinion the original writing was clear. However, as argued by Hladi\v{s} et al., M2OR contains screening data which is a source of considerable label noise and the test set of "M2OR" dataset should consist of ec50 data only. Therefore, in my understanding, both test sets of "M2OR" and "M2OR (EC50)" should contain only ec50 data and I would expect them to be drawn from the same distribution. Thus, the performance on these datasets should be comparable and the only difference between the "M2OR" and "M2OR (EC50)" are training sets, but test sets are drawn from the same pool of ec50 pairs. Could the authors clarify whether they used only ec50 pairs in "M2OR" evaluation? And if yes, then what is the difference in evaluation in Table 3 and Table 9?
> > >
> > > ## Q2
> > > At the end of the Section 4.2 the authors claim "LORAX is generating a unique representation, distinct from the original foundation model, that is better suited to describe odorant-receptor relationships and therefore generating a more informative odor space." While this is technically true, it is the primary goal of any fine-tuning procedure to obtain a representation that is aligned with the underlying task. Currently, the writing may suggest to the reader that the alignment to the neuronal representation is a unique feature of LORAX, but in the light of the authors' response that the result is unsurprising, what is the primary message and novelty of Section 4.2?
> > >
> > > ## Q3
> > > I thank the authors for clarifying the splitting strategy. While it seems fine, I still believe that adding experiments similar to the ones performed in Table 2 on other datasets would be valuable for the reader. In particular, a comparison to out-of-distribution performance reported by Hladi\v{s} et al. would give the reader a better understanding of the advantages and limitation of LORAX.

---

> > > > ### Author Response · Authors · 2025-11-25
> > > > **Response to Reviewer P2Me's Additional Questions**
> > > >
> > > > We thank the reviewer for their additional questions and clarifications. Please find our responses below.
> > > >
> > > > ## W1 and W3 of reviewer r6Qy
> > > > We would like to clarify that while numerous protein-molecule interaction models have been proposed since Hladiš et al., to our knowledge, none have reported results on the M2OR dataset. Thus, empirical evidence for their relative performance is currently lacking, and our study provides the first comparison on M2OR involving both the established SOTA (Hladiš et al.) and one of these more recent approaches (ProSmith). Furthermore, by leveraging LoRA-based approaches, our work improves upon both the established SOTA $\textit{and}$ a recent molecule-protein interaction model.
> > > >
> > > > We recognize the importance of extensive benchmarking. However, implementing, adapting, and evaluating all relevant recent models on M2OR is itself a substantial undertaking, beyond the scope of this work. Our work represents an important first step in this direction by establishing baseline performance. We hope this will facilitate further comprehensive benchmarking in future studies.
> > > >
> > > > On the point of statistical significance: following the reviewer’s suggestion, we conducted a paired t-test on the MCC values for LORAX and ProSmith in Table 3, obtaining a p-value of 0.08. While this does not meet the conventional threshold for significance, we note 1) our substantial gains over Hladiš et al., and 2) that the observed improvement over ProSmith is still suggestive of meaningful progress in the field.
> > > >
> > > > We hope the reviewer appreciates that our work lays necessary groundwork for broader benchmarking of next-generation models on M2OR, therefore making our work valuable to the community.
> > > >
> > > > ## Q1
> > > > We appreciate the reviewer’s careful attention to the composition of the M2OR datasets. To clarify:
> > > >
> > > > - For "M2OR", reviewer is correct in that the test set contains only EC50-labeled pairs, as per Hladiš et al., and the training and validation splits include primary and secondary screening data in addition to EC50.
> > > > - For "M2OR (EC50)", all splits (training, validation, and test) contain only EC50 data.
> > > >
> > > > However, while both dataset test-sets include exclusively EC50 data, because the test sets differ in both size and composition (e.g. the same test set for fold 1 of “M2OR” is not the same test set for fold 1 of “M2OR (EC50)”), the distributions are not identical. As a result, the evaluations presented in Table 3 and Table 10 (previously Table 9) are not directly comparable.
> > > >
> > > > ## Q2
> > > > We would like to reiterate that the main goal of LORAX, motivated by our observation of substantial overlap in foundation model feature spaces, is to introduce a tailored fine-tuning strategy that enhances the domain applicability of chemical foundation models for olfaction. While it is true that any fine-tuning procedure seeks to align representations with the underlying task, chemical foundation models have not previously been fine-tuned for odorant-receptor interaction prediction. Section 4.2 is intended to demonstrate LORAX’s capability to generate more meaningful representations for this domain, marking a key advantage over other models such as ProSmith, which retain unmodified representations throughout training.
> > > >
> > > > The insights provided by Section 4.2 are an essential part of the overall contribution of this work: establishing that LORAX is not merely another application of chemical foundation models for prediction, but rather a principled approach to fine-tuning molecular embeddings. This creates a new feature space that resolves representation alignment, an advance that has not previously been achieved in this domain.
> > > >
> > > > ## Q3
> > > > We thank the reviewer for this suggestion. We agree that extending the experiments in Table 2 to the other datasets in our work would provide valuable insight into the out-of-distribution performance of LORAX. We would like to note we are facing resource constraints given the number of additional experiments requested by other reviewers, and may not be able to include these results during the current discussion period. Nevertheless, if time allows, we will report these metrics for all relevant datasets in the final version of the paper.

---

> > > > > ### Comment · Reviewer_P2Me · 2025-11-26
> > > > >
> > > > > ## Q1
> > > > > I appreciate the authors' engagement, and I believe that this is a crucial point that needs clarification, as it may suggest a fundamental misalignment in the presented results. As written in line 175, M2OR (EC50) is evaluated on 5-fold cross-validation with random (uniform?) splits. Considering that for Hladis et al., the values are taken from the original publication (see line 398), then I assume that M2OR test set is sampled uniformly from the set of ec50 pairs as done by Hladis et al. If this is the case, then both M2OR (EC50) and M2OR test sets are uniformly sampled from the same set of values, thus they are indeed identically distributed. Therefore, even if the number of samples differ, the mean performance across 5 runs should be comparable to some extent. Following this observation, I find the fact that according to Table 10, ProSmith model with random features have nearly the same performance as ProSmith in Table 3 surprising and missing an explanation.
> > > > > It is possible that I missed some point, but as it is written in the manuscript, I find the results unclear, and I think that a clarification is needed for this point.
> > > > >
> > > > > ## Q2
> > > > > "LORAX is not merely another application of chemical foundation models for prediction, but rather a principled approach to fine-tuning molecular embeddings."
> > > > > In my understanding, as LORAX is simply a combination of ProSmith and LoRA, then in my opinion, the results show that "LoRA is a principled approach to fine-tuning (molecular) embeddings" which is a well established result and the main reason for the popularity of LoRA.
> > > > >
> > > > > ## Q3
> > > > > While I understand the difficulty to run many experiments in the short time period of the rebuttal, I tend to not consider this as a valid reason for the acceptance of an incomplete work. Ultimately, scientific publication at a venue such as ICLR should provide a novel and complete piece of knowledge for the reader and if an important set of experiments can not be run during the rebuttal period, then the authors may consider submitting the completed work at a subsequent venue.
> > > > >
> > > > > Overall, I think that the manuscript was not ready for the submission. The novelty of LORAX is limited, since it is a combination of an already existing model with a widely used LoRA framework and the main contribution of the paper tends to be in a direction of a benchmark paper. However, for a benchmark paper I find several experiments to be missing (e.g. out-of-distribution evaluation on all datasets, evaluation of other DTI models) and I do not think that there is enough time to finish these experiments during the short rebuttal period. Nonetheless, when additional experiments and further analysis/interpretation of the surprising alignment between molecular representations are added to the paper, the work can be resubmitted to a next conference.

---

> > > > > > ### Author Response · Authors · 2025-11-29
> > > > > > **Response to Official Comment by Reviewer P2Me**
> > > > > >
> > > > > > We thank the reviewer for their response.
> > > > > >
> > > > > > ## Q1
> > > > > > We examined the distributions of responses across all splits for both the M2OR and M2OR (EC50) datasets (see Supplementary Figure 3 for an example from the first split). While the overall distributions are similar, the actual data in each fold are different, with minimal overlap, as shown in the newly updated supplement and described earlier. As a result, differences between the prediction metrics reported in Table 3 and Table 10 arise from differences in training data composition and the random assignment of protein-molecule pairs to test sets. This is expected, since the models are trained on fundamentally distinct datasets and will naturally demonstrate variations in performance. We have added text to Table 10 to note this, to minimize confusion, but only in Table 10 (not Table 3) because we only report metrics for this data in the appendix and we don’t want to introduce confusion in the main text. Importantly, these differences, perhaps as the reviewer feared, do not affect the broader conclusions of our paper. The two datasets are distinct, and direct comparison of their respective performance is not meaningful, contrary to the reviewer’s suggestion. We believe that model strength is best assessed on the full M2OR dataset, which is presented in the main text and which is the standard benchmarked by other models such as Hladiš et al., and on which our approach demonstrates SOTA performance. After careful consideration, we do not find sufficient evidence to support the reviewer’s claim of a “fundamental misalignment”. We hope that the added text and rewording of the paper to focus on the full M2OR dataset (to accommodate requests by other reviewers) will reduce the confusion the reviewer experienced, and believe with the modified text, this rather minor point will not be of concern for most readers.
> > > > > >
> > > > > > ## Q2
> > > > > > We respectfully disagree that our work is a straightforward combination of ProSmith and LoRA or that its contribution is limited to reaffirming LoRA. The core novelty and motivation of our work lies in addressing the domain applicability of LoRA within the context of chemical foundation models, specifically for tasks in olfaction, a setting that, to our knowledge, has not been studied in prior literature. This is further supported by multiple reviewers who highlighted our approach as both novel and impactful for this very reason. We believe reducing our method to a combination of existing methods and dismissing its novelty is an unfair assessment. We reiterate: There is, to our (and multiple reviewers) knowledge, no existing work that implements our approach. We have added text to the main results of our paper further highlighting this point, both to make this more clear to readers who might not appreciate this point as a strength (as other reviewers have recommended) and to draw attention to this important contribution for other readers who may not think it sufficiently impactful without additional context that we will provide.
> > > > > >
> > > > > > ## Q3
> > > > > > We, as a courtesy, let the reviewer know that we may not have time to get to this experiment but we are still endeavoring to complete them. There is about a week before the discussion period ends, and we try our best to get to these experiments in the limited time remaining!
> > > > > >
> > > > > > We thank the reviewer for their time and effort in evaluating our submission. While we value critical feedback and are committed to improving our work, we are surprised by the critical tone of the review, as it was not accompanied by specific, actionable recommendations to address perceived shortcomings. Regarding the suggestion to evaluate additional unspecified DTI models, we believe this falls outside the scope of our work, which already presents a comprehensive set of findings: noting an interesting overlap in chemical foundation model feature spaces and proposing a model which we then benchmark against multiple SOTA approaches. Notably, to the best of our knowledge, our work is the $\textit{first}$ to implement and evaluate new DTI models on the M2OR dataset, thus establishing a strong foundation for future benchmarking. In light of the strengths highlighted by other reviewers and addressed in our earlier responses, we feel that the request for additional benchmarking is not reasonable in the context of this submission. We are grateful for the reviewer’s insights and hope the current submission, along with our responses, convincingly demonstrates the value of our contributions.

---

> ### Author Response · Authors · 2025-12-03
> **Completed Statistical Testing and Generalization Studies**
>
> We would like to let the reviewer know that we have completed all statistical testing and generalization studies suggested by the reviewer. For the statistical testing asked for by the reviewer, please see response "Completed Statistical Tests and a Thank you to the Reviewer" to reviewer Cjx2. Generalization studies for both unseen odorants and unseen receptors were run for the Hallem dataset and the M2OR (EC50) dataset and provided in the supplement in Table 9. We note here that the reviewers claim of not doing experiements within the rebuttal time frame and threfore "not consider[ing] this as a valid reason for the acceptance of an incomplete work", is now moot, since we have addressed every actionable request specified by the reviewer.

---

### Official Review · Reviewer_r6QY · 2025-10-28

**Soundness:** 2
**Presentation:** 3
**Contribution:** 2
**Rating:** 4
**Confidence:** 3

**Summary:**

In this paper, the authors contribute LORAX, a novel method that incorporates fine-tuned chemical foundation models and pre-trained protein foundation models for odorant-receptor binding prediction tasks. The authors motivate the need for their approach with an extensive evaluation of different representations across relevant datasets, highlighting: (i) that the representations encoded by pre-trained chemical foundation models do not significantly improve the performance in odorant-receptor binding prediction tasks versus hand-crafted physiochemical features; and (ii) that is due to the significant alignment between these pre-trained representations and the hand-crafted ones. Finally, the authors evaluate LORAX across different datasets, showing that it learns a more generalizable representation to unseen odorants, and provides a representation space more similar to the underlying neural representation.

**Strengths:**

- **Originality**: The originality of this work is highlighted in two different aspects: (i) the use of low-rank adaptations to fine-tune foundation models of chemical data for olfactory tasks; (ii) the use of such fine-tuned models in combination with pre-trained representation models of protein data for odorant-receptor binding prediction tasks. While fine-tuning foundation models of chemical data for olfactory tasks has already been explored before (see [1]), the use of LoRAs for this purpose is, to the best of my knowledge, novel.

- **Quality**: I found the paper to be of good quality, with minimal typos (one exception being "adapated" in line 435). Despite this, I encourage the authors to improve the quality of some of the figures (as an example, Figure 1B (a) is not really comprehensible without improving the caption). While not particularly novel, the idea of using low-rank representations to fine-tune foundation models is sound.

- **Clarity**: The paper is overall clear. In particular, I thoroughly enjoyed the structure of the work, starting by motivating the need for LORAX with extensive experimental results, and subsequently evaluating LORAX, comparing to previous results.

- **Significance**: This work contributes to an emerging line of research within olfactory machine learning that resorts to pre-trained foundation models of chemical data, due to the lack of large-scale, high-quality and diverse datasets of olfactory data. Fine-tuning approaches, such as the one presented in this paper, are undoubtedly an important tool to address the specificity of olfactory data, and thus significant.

**References**:
- [1] Taleb, Farzaneh, et al. "Can transformers smell like humans?." Advances in Neural Information Processing Systems 37 (2024): 72032-72060.

**Weaknesses:**

- In Section 3, the authors present an extensive study on affinity prediction, considering different odorant representation models and input conditions: only odorant, molecule + odorant, and a processing of these two representations using ProSmith. The results show a significant increase when using protein information and ProSmith. However, I've found interesting that no ablation condition with only protein information was done, especially given that this information seems fundamental for the task itself. Can the authors show results of such ablation?

- It is not clear what are the main differences between LORAX and ProSmith, beyond the LoRA component of the chemical model. Can the authors elaborate on the main differences between the two models?

- The authors motivate LORAX by highlighting that current odorant representations perform similarly to each other, yet the results in Line 317 show no significant performance difference between LORAX and ProSmith. Same for the results in Table 3, where ProSmith actually achieves a significantly higher AUROC. While achieving SOTA is not a requirement across all metrics and situations, in this case it is unusual since the authors are fine-tuning the pre-trained chemical representation for their specific downstream evaluation task. Can the authors comment on this? Why would a practitioner use your approach instead of ProSmith?

- Since LoRAs is such an important component of their contribution (and the main difference between their approach and the baselines), it is unfortunate that no extensive evaluation of different LoRAs is made, in regards to rank, layers to be fine-tuned, etc... Can the authors present this results? How did the authors select their LoRA architecture?

- Some important details are currently missing in the paper such as definition of the evaluation metrics in the tables, and identification of which datasets are used for the results presented in the tables. Can the authors add this information?

**Questions:**

See Weaknesses.

---

> ### Author Response · Authors · 2025-11-15
> **Response to reviewer r6QY's initial review**
>
> Thank you for the reviewers comments and suggestions. We appreciate the noted strengths indicating strong originality, quality, clarity, and significance. Highlighting that we are the first work to use LoRA in olfaction, we agree, is a substantial contribution. Given LoRA’s impressive performance in other domains, we found this gap in the literature surprising, and we believe our contribution represents work of significance that many in the ICLR community will appreciate seeing. We appreciate notes on clarity and the quality of the paper and agree that we present our work in a clear and structured way, motivating the need for our approach. We agree with the reviewer that our work addresses a key challenge in olfactory machine learning. We agree that leveraging foundation models in a field that lacks large-scale, high-quality, and diverse data is undoubtedly an important and significant addition to this field. With these strengths in mind, and our rebuttal (below) to weaknesses, we hope the reviewer will revisit their score.
>
> # W1: No “Protein Only” Model
>
> We appreciate the reviewer bringing this up! We created a model (Protein only, PO) that only uses the protein (see Table 2 in supplementary file). This model performs similarly to the MO model, highlighting the advantage of incorporating both protein and molecular information, as we note in the main text.
>
> # W2: Difference between LORAX and ProSmith
>
> The main difference between LORAX and ProSmith is the LoRA adapter added to the chemical foundation model. This allows the model to adapt the molecular representation over training, leading to increased performance and superior molecular representation, outlined in the text. Additionally, ProSmith was not created for olfactory datasets, whereas LORAX was developed for olfaction (e.g. molecular representation generation).
>
> # W3: Limited Performance Gains of LORAX Over ProSmith
>
> While our model does not surpass SOTA on every metric, please see our arguments in “Weakness: Metric choice and class imbalance” in response to reviewer Cjx2, which explain why MCC and F-score are especially informative for the M2OR dataset. In summary, MCC and F-score, two metrics in which LORAX excels over ProSmith, are important for datasets with heavy class imbalance like M2OR.
>
> We also point out that the previous SOTA model on M2OR, is not ProSmith, but rather the model in Hladiš et al., against which we show drastic improvement. Though we do not have access to their splits (and thus cannot perform statistical testing), we show that our model is superior. Had we simply omitted ProSmith in our analysis, for which at the time there was no precedent on these data, we would have easily demonstrated significant improvements. In other words, in our opinion, considering ProSmith as a SOTA to compare against seems unreasonable. Nevertheless, we wanted to demonstrate that ProSmith, which has demonstrated strong performance in a related domain (enzyme-substrate binding), shows strong performance on these datasets, and view this finding as a contribution in and of itself. On top of that, our results show that our model surpasses ProSmith, an approach not previously evaluated in the context of olfaction, further highlighting its effectiveness.
>
> Regarding the Carey dataset, although not statistically significant, we do see improvements over ProSmith (which, again, was not previously tested on these data). Due to the data’s limited size, the improvements are less pronounced. Additionally, in Table 2, LORAX demonstrates almost statistically significant improvement for unseen odorants compared to ProSmith, addressing a key challenge in olfaction and small molecule/protein binding. Models that lack generalization are only useful for compounds similar to those seen in training, limiting their impact on analysis of novel molecules. Lastly, there is interest in olfaction for generating molecular features tailored to olfactory tasks. LORAX creates representations that are specifically adapted to olfaction, offering a promising toolkit for developing olfaction-optimized molecular representations.
>
> # W4: Choice of LoRA parameters
>
> Our initial choices were guided by recommendations in the LoRA paper (Hu et al., 2021). We recognize the importance of more detailed exploration, and, in response to your insight, we will: 1) perform a systematic sweep over the rank of adapter matrices, and 2) evaluate performance of adapters on different combinations of query, key, and value matrices. We note these experiments require substantial compute and time due to the extended runtimes, but agree these investigations would significantly strengthen our work and its conclusions. We appreciate your suggestion and are committed to these analyses.
>
> # W5: Caption information
>
> Thank you for bringing this to our attention. We have added more information to the captions for many tables. If further clarification is needed, we will refine our captions to ensure greater clarity!

---

> ### Author Response · Authors · 2025-11-21
> **Completion of LoRA Hyperparameter Sweep**
>
> We would like to update the reviewer with our progress on W4, where we have completed a comprehensive hyperparameter sweep over both rank and adapted modules of our LoRA adapter. Specifically, we retrained our model on the Carey dataset using a variety of rank constraints for the LoRA module, as well as different combinations of adapted weight matrices. The results from these experiments have been added to our supplement (Table 3 and Table 4).
>
> Our analysis reveals very little variance in performance gains across both the different module combinations tested and the range of ranks explored. Notably, we observed a slight increase in performance with lower rank adaptation (r=2). This finding aligns with the original observations by Hu et al., in which even very low-rank adapter matrices are sufficient to effectively fine-tune models for new tasks.
>
> Although there are interesting trends in some of these results, the differences in performance remain modest overall. Based on this comprehensive sweep, we conclude that our default parameter choices used in the paper (rank=8, modules=[query, key, value]) are sufficient to demonstrate the effectiveness of our method. Our selected configuration provides a good balance between efficiency and performance without requiring exhaustive hyperparameter optimization.
>
> We appreciate the reviewer’s suggestion to further examine these parameters, as it has allowed us to rigorously validate the stability and generalizability of our approach. We hope this additional analysis satisfactorily addresses the reviewer's noted weakness. If the reviewer has any further considerations please let us know!
>
> Hu, E. J. et al. LoRA: Low-Rank Adaptation of Large Language Models. Preprint at https://doi.org/10.48550/arXiv.2106.09685 (2021).

---

### Official Review · Reviewer_Cjx2 · 2025-10-31

**Soundness:** 3
**Presentation:** 4
**Contribution:** 2
**Rating:** 4
**Confidence:** 4

**Summary:**

In this work, the Authors investigate the representations of odorants through the lens of their predictive power for the odorants' affinitie for olfactory receptors. To this end, they compare standard sets of chemical features and several representations learned in foundation models for chemicals and proteins. Upon finding the overall similarity between different representations of odorants, they propose a LoRA-based finetuning framework for learning the representations relevant to olfaction. They test treir and existing representations on several odorant-receptor affinity datasets.

**Strengths:**

- Comprehensive literature overview

The paper considers a comprehensive set of the representations (e.g. the chemical descriptors) and the models (e.g. GNNs) that have found widespread use in olfaction; thus, the work is well-positioned in literature and relevant to the olfactory community

- Comprehensive evaluation

For both existing and new representations, the work offers a comprehensive analysis, thus adding new knowledge about the tools that people use in practice and not only about the new method.

- Insights into the similarity and differences between conventional representations

They find, through the analysis, that many different representations of odorant are, in fact, similas, especially those leading to good predictions for odorant-receptor affinities. Additionally, the similarity between the decodeability is shown for the most of the representations

- Ablations

Futhermore, the imapct of different model inputs is investigated; the utility of the joint condiferation of the representations of many odorants and many reptors in a single model (e.g. ProSmith) is shown.

- Text clarity

The text is super clear and well-structured.

**Weaknesses:**

- The proposed model doesn't seem to improve upon the baselines.

Through Table 2, LORAX is better than ProSmith for unseen odorants but worse for unseen receptors (that said, all R2-scores here are really small). Through Table 3, LORAX is the best in recall and F-score but not the best in precision and AUROC. Unless there's a principled way to determine that these metrics are more important thatn the others, one can't claim that the model is the best.

**Questions:**

What substantiates that claim that LORAX has the best representations?

---

> ### Author Response · Authors · 2025-11-13
> **Response to reviewer Cjx2's initial review**
>
> We thank the reviewer for their response. We especially appreciate all the noted strengths of our paper: comprehensive literature review and evaluation, insights into various chemical representations, ablation studies, and clarity. We agree with the reviewer that these represent some significant strengths of our work, and we are glad that came through! Below we address the reviewers weaknesses and questions. We admit we were surprised by the discrepancy in our score and the limited number of weaknesses supplied by the reviewer. Given the many noted strengths, along with the weaknesses we respond to here, we hope the reviewer will consider revisiting their score.
>
> $\underline{\text\{Weakness: Baseline improvement}}$
>
> We appreciate the reviewer’s concern regarding LORAX’s improvements over existing methods. While the overall performance gains of LORAX may appear modest, the model exhibits notable improvements in generalizing to unseen odorants (Table 2). This is a difficult challenge even for SOTA approaches such as ProSmith, which claim performance increases in this scenario [1]. Additionally, LORAX demonstrates significant gains in MCC and F-score on the M2OR dataset (Table 3), two metrics that are particularly essential given the dataset’s pronounced class imbalance (see next point). These advances, though incremental, are meaningful, and we believe they are important to highlight.
>
> We would also like to note that the LORAX model contains other features, outside of improving predictive score, that indicate improvements over existing methods. Our model introduces the ability to adapt chemical foundation model representations via LoRA adapters, yielding unique molecular representations. Figure 4 demonstrates that these representations are more effective for downstream prediction than others tested, while Figure 5 shows their improved alignment with neural activity. These characteristics represent important methodological advances beyond general performance increases.
>
> $\underline{\text\{Weakness: Metric choice and class imbalance}}$
>
> Here we make an argument for why MCC and F-score are more robust metrics for assessing performance for M2OR.  The M2OR dataset is highly imbalanced (~6.6% responsive class), making AUROC less reliable indicators of true model performance [2]. Metrics like MCC and F-score, which better account for class imbalances [3,4], are thus more appropriate and more reliable. Taking into account the pronounced class imbalance in the dataset and LORAX’s superior performance in MCC and F-score, we believe that LORAX is performing better than ProSmith on the M2OR dataset. From your insight, we have added clarifying text to the manuscript (lines 382-387) to highlight this point! We have also added a supplementary table (Table 1, Supplementary Material) highlighting the class imbalance in the data.
>
> $\underline{\text{Question 1}}$
>
> We believe that LORAX generates molecular representations uniquely suited for odorant-olfactory receptor binding prediction, as supported by results in Figures 4-5 and Tables 2-3. In Figure 4, LORAX shows notably higher validation performance than ProSmith, and the XGBoost ensemble analysis highlights that the <cls> token LORAX generates is much more informative than ProSmith's. Since the main architectural difference between ProSmith and LORAX is the LoRA adapter on the chemical foundation model, we attribute these gains to LoRA-enabled, task-adaptived representations. This suggests more effective feature extraction by LORAX's transformer. Figure 5 further demonstrates that LORAX learns representations of odorants that are more closely related to neural representation over training. ProSmith lacks the ability to actively adapt these representations, meaning the molecular feature will be constant throughout training, and less aligned with neural activity.
>
> These figures are explanations to why we see improved performance with LORAX over ProSmith and other SOTA methods in Tables 2 and 3. We believe that the ability to actively adapt the representation of the chemical foundation model is a crucial component to generate better generalization capabilities (Table 2) and better performance metrics on data (Table 3).
>
> $\underline{\text\{Final Thoughts}}$
>
> We believe that we have answered the reviewers questions and addressed the reviewers proposed weaknesses in full. If further support is needed to demonstrate the significant contribution our work has to the ICLR community, we are happy to engage in conversation about those concerns. We believe the reviewer brings compelling points, and we have changed our paper accordingly to address them. If there are still concerns, please reach out again and we will be happy to address anything further!
>
> Ref:
> [1] Kroll, A. et al. PLOS Computational Biology 20, e1012100 (2024).
> [2] Davis, J. & Goadrich, M. ICML (2006).
> [3] Chicco, D. BioData Min 10, 35 (2017).
> [4] Chicco, D. & Jurman, G. BMC Genomics 21, 6 (2020).

---

> > ### Comment · Reviewer_Cjx2 · 2025-11-22
> > **Thank you**
> >
> > With this, I'd like to thank the Authors for their responses and their engagement with the rebuttal process.
> >
> > I have carefuly read the response here, other reviews, and the responses to other reviews. I summarize my thoughts below.
> >
> > First, regarding my original score: While, as I have mentionined in my original review, I see mulptiple strengths of this work, the final arbiter of an ML model is its performance on the data; ideally, on the data already used by other people. The empirical results provided in this paper, in the Authors' own words, appeared modest. Hence my rating.
> >
> > Second, I appreciate the Authors' response that highlights _other_ strengths of the paper. I list these below:
> >
> > - Prioritizing the MCC and F1-score because of the class imbalance. Thank you for highlighting that (including in the text); this is a valid point. While I'd like to debate the claim regarding the strength of the results in Table 2 (the R2 of 0.04 +- 0.6 is _very_ small and, judging by the provided numbers, not significantly different from zero, i.e. the chance decoder) and while I'd like to highlight the scale of the the improvenents in Table 3 (2% in F-score and 1% in MCC; significance unclear provided the standard deviations of 3% and 4% respectively), at least this is a substantiated argument.
> >
> > - The results in Figures 4 and 5 are important. Thanks for bringing my attention to that. As these may be the strongest results of the paper, it may make sense to put them earlier in the text so that the reader's focus is guided appropriately. In Figure 4, we see that LORAX pay attention to the learned features as opposed to the input features (although that doesn't increase the decodidng accuracy by much). In Figure 5, we see that LORAX's representations are most aligned to neural ones (although I'n not sure how to gauge these numbers, i.e. what are the refernce distances for "good" or "bad" similarity, what makes a meaningful improvement, and what are the error bars for these values (e.g. through bootstrap)).
> >
> > Overall, it looks like the paper has a few interesting leads that, however, could use different levels of future development (some of them are simple!)
> >
> > In my opinion, to qualify for the borderline accept, the following additions are necessary:
> > - For Table 2, reporting the t-tests w.r.t. chance OR arguing why the performance non-distinct form chance is still a valuable result;
> > - For Table 3, reporting the t-tests (or the paired sign tests) on the differences with, e.g., the Benjamini-Hochberg false discovery rate (FDR) correction to establish the significance of the improvements;
> > - For Figure 5, reporting, e.g., bootstrap + FDR to see what differences are significant and providing reference values for "good" and "bad" distances;
> > - (Optional) a discussion of where the similarity of the leaned representations and the neural activity could be helpful.
> >
> > Higher scores, in my opinion, would rerequire large increases in model's performance on established datasets (potentially with the use of additional data, as we speak of foundation models here). I do acknowledge, however, that this is diffucult due to the limited availability of datasets in olfaction.

---

> > > ### Author Response · Authors · 2025-11-24
> > > **Response to reviewer Cjx2's Comment**
> > >
> > > We would like to thank the reviewer for the positive and constructive feedback, as well as for providing specific recommendations regarding statistical analyses! The reviewer's guidance is greatly appreciated and will be very helpful for effectively presenting the strengths of our model. We would like to note that, unfortunately, we do not have access to per-split performance metrics from the Hladiš et al. model (Table 3), which prevents us from conducting a direct paired t-test with those results. Nevertheless, we will incorporate the suggested statistical tests into the paper wherever possible. Before doing so, however, we would like to emphasize an important point to address the reviewers concerns below.
> > >
> > > We agree, as the reviewer noted, that the final arbiter of an ML model is its performance on the data used by other people, and that a large increase in model performance on established datasets is the strongest measure of strength. In light of the reviewers response, we’d like to highlight a conversation we’ve been having with another reviewer (see “W3: Limited Performance Gains of LORAX Over ProSmith” with reviewer r6QY) which we think is pertinent here, and will help us understand how we can satisfy this reviewer’s critiques in a way that is fair and also reflective of this rapidly evolving field. These were not points we made strongly in our initial response, but have come to the surface with other reviewers during review.
> > >
> > > ​​Specifically, while much of our comparison revolves around ProSmith, it is worth clarifying that the current SOTA for the M2OR dataset is actually the model proposed by Hladiš et al. Our results demonstrate significant gains over this benchmark, which we view as the most reasonable standard for comparison. Hladiš et al. is the most recent peer-reviewed model that has been systematically evaluated on M2OR; therefore, if the requirements for a strong contribution is a comparison of an established model on an established dataset that others use, by comparing LORAX directly to Hladiš et al., we satisfy the established criteria for demonstrating a strong contribution with respect to recognized baselines and datasets in the field. ProSmith, prior to our work, has not demonstrated strong performance on M2OR, and we are the $\text{\textit{first}}$ group to show performance using ProSmith on M2OR. So while our improvements over ProSmith are modest, the more meaningful metric is our advancement over the established SOTA (i.e. Hladiš et al.). We recognize that using ProSmith as a baseline without appropriately emphasizing this context could inadvertently understate the substantive progress our method represents.
> > >
> > > We plan to clarify this distinction in the manuscript and to ensure our results are presented in a way that highlights both incremental gains and substantive advances over recognized SOTA models. We want to emphasize that we will not ignore the modest gains LORAX has over ProSmith, but rather reframe our language to emphasize comparisons to other relevant methods, including those most established in the field.
> > >
> > > To this end, we would like to ask the reviewer for clarification: given that Hladiš et al. represents the current SOTA on the M2OR dataset, would demonstrating substantial performance gains over this model, together with the recommended statistical analyses wherever the data allow (such as with ProSmith), be sufficient for a fair and comprehensive assessment of our model? We will also continue to report improvements over ProSmith for completeness, but would appreciate the reviewers guidance on whether establishing significant progress over Hladiš et al. alone adequately addresses the reviewers criteria for a strong contribution.
> > >
> > > Thank you once again for your careful reading and constructive input!

---

> > > > ### Comment · Reviewer_Cjx2 · 2025-11-27
> > > > **Re:**
> > > >
> > > > Thank you for your timely and detailed response.
> > > >
> > > > Regarding the specific tests: I take your point that the comparison with Hladis et al' s work would require per-split numbers that you don't have access to. Let's not do it now but for the final version of the paper I'd suggest either reaching out to the authors of that work or reproducing their experiments to obtain the numbers.
> > > >
> > > > As for the other tests (comparisons with ProSmith on Tables 2 and 3) and bootstrap for Figure 5, can those be done? I think, as long as you've brought up ProSmith, it would be interesting and important for the community to see the significance of the differences.
> > > >
> > > > Regarding cuurent SOTA, I take your point as well. As the numbers for ProSmith on M2OR are actually **your** results, this needs to be highlighted (it turns a seeming weakness into a strength of your work) and you also don't have to be better than them to fulfill the ICLR's criteria. But please, kindly, do the statistical significance tests for the comparisons with them. That would be super useful for the commiunity even if your model won't always be the best (also personally curious to see the outcomes to guide some of mine research).
> > > >
> > > > Overall, in light of this response, I'm raising my score.

---

> > > > > ### Author Response · Authors · 2025-11-28
> > > > > **Completed Statistical Tests and a Thank you to the Reviewer**
> > > > >
> > > > > We would like to update the reviewer that we have completed all statistical tests the reviewer has recommended. Below we detail the completed tests and insights we get from them.
> > > > >
> > > > > ## Table 2 reporting metrics w.r.t. chance
> > > > > As suggested, we have added a "Naïve" model to Table 2, which predicts the mean value of the training set for all test set entries. We conducted a non-parametric repeated measures ANOVA (Friedman test) and report p-values with FDR correction in Supplementary Tables 5 and 6. These results show that LORAX is very near the threshold for statistical significance against the Naïve model in both scenarios ($p=0.069$), and is statistically comparable to ProSmith in the unseen receptor scenario. This further strengthens our claims of LORAX's capabillity to improve over existing models in two common generalization scenarios!
> > > > >
> > > > > ## Table 3 reporting paired t-tests
> > > > > We report another Friedman test FDR multiple comparison corrections for Table 3 in supplementary Table 7. We included MolOR in this assessment because, as per the request of reviewer jDeV, we retrained the MolOR model from scratch to report all metrics for direct comparison. We see that LORAX significantly outperforms both MolOR (p-value<0.003) and approaches significantly outperfirming ProSmith (p-value=0.067) from these tests. This further confirms the ability of LORAX to out perform both SOTA models in the field $\textit{and}$ SOTA protein-molecule interaction models never before tested on this dataset!
> > > > >
> > > > > ## Bootstrap Figure 5
> > > > > We added a bootstrap analysis (Figure 2 in the supplement) for Figure 5. We perform an ANOVA test with an FDR corrected multiple comparissons test and show that almost all differences between the initial representation (ChemBERTa-77M-MTR) and the final representation (LORAX) are statistically significant.
> > > > >
> > > > > We thank the reviewer for their careful feedback and their intention to raise their score! We are grateful for the recommended statistical tests, which have further strengthened the impact and rigor of our work.

---

### Official Review · Reviewer_jDeV · 2025-11-01

**Soundness:** 3
**Presentation:** 3
**Contribution:** 4
**Rating:** 6
**Confidence:** 4

**Summary:**

The authors show that for olfactory tasks, chemical foundation models do not yield representations that outperform physicochemical descriptors on odorant-receptor binding tasks. To bridge this issue, they propose a LoRA strategy to finetune these representations, achieving better predictive performance and generalizability over their benchmarks. The idea is clearly communicated and straightforward to understand, and the authors do reasonably well in their evaluation of existing foundation models to emphasize their point. However, in my opinion, the conclusion could be made stronger with the inclusion of a few more key benchmarks that would make the results more convincing. I recommend that the paper be accepted to ICLR after these changes are satisfactorily addressed.

**Strengths:**

The authors present the idea in a clear manner -- there are varying approaches towards the representation of molecules in olfaction, and the lack of olfactory data can make the use of foundational models more desirable over a supervised learning approach. There needs to be a benchmark for the efficacy of foundational models towards olfactory tasks, and this work addresses this gap.

The authors also evaluate a wide variety of representations beyond foundational models, and consider supervised learning approaches in addition to physicochemical representations. I appreciate that their analysis involves comparisons of performance spreads beyond just looking at mean performance.

I particularly enjoyed the analysis in Sec 4.1 where the authors analyzed feature importance to detect that the ProSmith model intentionally underutilizes the transformer feature.

**Weaknesses:**

Table 3 is incomplete in my opinion -- to fully compare everything on equal footing, the authors should use the approaches Hladiš et al. and Chithrananda et al. used in their paper to complete the entirety of Table 3. In line with this comparison, I would like to see the performance of a supervised GCN (widely used in other olfactory tasks) on all the datasets to properly show that fine-tuned foundational models indeed provide a better representation.

Additionally, I concur with the authors in Appendix A that they’ve evaluated a large number of foundation models, but as they also mention, these models are not recent ones, and a conclusion like the authors are proposing needs to be based on models that are more recent.

I was partially confused by Figure 4B, and thought a more natural way to show that LORAX is useful is to show that the proportion placed on < cls > is higher for LORAX to be consistent with their analysis with ProSmith’s poorer performance compared to LORAX.

I think the authors should dial back their claims that this phenomenon is general for all olfactory tasks, as they have only demonstrated this for molecule-receptor binding tasks.

The authors propose a strategy where they concatenate the features, but cross-attention has also proven to be a reasonable strategy in molecule-receptor binding tasks (i.e. Chithrananda et al.).

**Questions:**

1) I recall that the M2OR dataset is very sparse in terms of molecular-receptor pairs, and so it should not be surprising that the MO model performs poorly due to the lack of data. I’m not sure if this is the case for the other two datasets the authors use in this work.
2) What is the source of the neuron response data that is used to generate the neuron representation in Figure 5?
3) What is the performance of a GCN supervised baseline?
4) What is the performance of a more recent foundational model?
5) Does cross-attention outperform concatenation, and does it lead to different conclusions from what you observe?

---

> ### Author Response · Authors · 2025-11-17
> **Response to reviewer jDeV's initial review**
>
> Firstly, we would like to thank the reviewer for their in depth review and their careful read through of our work. We greatly appreciate the reviewers detailed comments regarding the clarity of our work and the importance of rigorously assessing the efficacy of chemical foundation models within olfaction. We fully agree that a comprehensive comparison between foundation models and classical molecular descriptors has been notably absent in the existing literature, and our work aims to address this important gap. We are also grateful for the reviewer's recognition of our analysis in Section 4.1, which we believe demonstrates the effectiveness of our approach and underscores LORAX’s potential as a valuable contribution to the broader literature. We appreciate and agree with the reviewer noting that our work is a strong contribution to machine learning in olfaction! We hope that, given our responses and revisions below, the reviewer may reconsider their score.
>
> # W1: Incomplete table 3
>
> We note that both Chithrananda et. al. and Hladiš et al. use a GNN, in combination with a protein foundation model, to perform prediction. Additionally, our model can be directly compared with Hladiš et al., as we use the exact same data splits and data weighting. Given that both these models use a GNN and a protein foundation model, is this sufficient to prove our method does better than a supervised GCN baseline? Or would the reviewer still want us to train both methods from scratch?
>
> # W2: More recent models
>
> We appreciate that the reviewer addressed this point and noted that we explain this in the appendix. We want to reiterate that the field of chemical foundation modelling is rapidly evolving making it difficult to incorporate the newest method. There are some pitfalls for incorporating new methods: 1) LoRA was originally developed for transformer architectures and parameterizations vary significantly across different GNN implementations, making it harder to generalize where and how LoRA can be applied. This makes many of the new GNN-based foundation models like Uni-Mol and MolGPS difficult to incorporate into our framework. 2) Many new models are not on Huggingface (e.g., MolGPS does not have a publicly available github to our knowledge) and are custom designs that cannot easily be used with a LoRA module. For this reason, we believe it will take some time to test out a different state of the art chemical foundation model. This being said, we will work on incorporating Uni-Mol (Zhou et al., 2022) into LORAX and report performance improvements as soon as possible. Again, this may take some time, but we will try to build a version of LORAX that incorporates this newer GNN foundation model by the time discussion closes. Would this be sufficient to address the reviewers' concerns and, in the reviewers opinion, make the paper a stronger contribution to the ICLR community?
>
> # W3: XGBoost weighting in Fig4B
>
> We appreciate this point that the reviewer made here, and we think the reviewer is correct that there is a more natural way to highlight this point. We have retrained the XGBoost ensemble for every example using only two XGBoost models: one using the <cls> token only, and one using the original chemical and protein representations. We have added this figure to the supplementary file (Figure 1). This confirms our original findings, showing the LORAX <cls> token is weighted more heavily than the other representations.
>
> # W4: Dial back claims
>
> We appreciate this point and will work to make the language less grandiose. For clarification, are there specific passages that the reviewer believes are in need of alteration?
>
> # W5: Feature concatenation
>
> We believe there is a slight misunderstanding either on our part, or the reviewers' part, here. We do utilize a cross attention block in the model (Figure 3A, first yellow box). Let us know if we are misunderstanding each other and the reviewer is thinking of concatenation elsewhere in the model!
>
> # Q1
>
> The reviewer is correct in noting that M2OR is sparse in terms of pairs. This is why we exclude receptors with fewer than 50 entries when training the MO model. The other two datasets, however, contain results from every odorant-receptor pair. We see poor performance across each dataset with the MO model regardless of the sparsity of the data.
>
> # Q2
>
> The ‘neural_representation’ in Figure 5 is generated by creating a feature vector that is composed of every receptor response. For example, if odorant 1 activates receptor 1 with 2.5 spikes/s, receptor 2 with 12.1 spikes/s, and receptor 3 with 1.11 spikes/s; the feature vector of odorant 1 would be [2.5, 12.1, 1.11]. We then do CCA analysis on the similarity of all odorants featurized in this way vs. LORAX and the original ChemBERTa-77M-MTR feature from which it was fine-tuned.
>
> # Q3
>
> Please see response to W1
>
> # Q4
>
> Please see response to W2
>
> # Q5
>
> Please see response to W5

---

> > ### Comment · Reviewer_jDeV · 2025-11-17
> >
> > Thanks for your engagement in the peer review process. Here are my thoughts:
> >
> > W1: I was specifically referring to populating Table 3 with Hladiš et al.'s AUROC, and Chithrananda et al's AveP, Precision, Recall, F-score and MCC -- these are not values that are present in their respective works, and require that you use their model to evaluate these same metrics. Apologies for being unclear before, but my request for the GCN baseline was for Table 1 and Appendix E, where a GCN trained from scratch would be compared against molecular descriptors and pre-trained models. A stronger claim could be made in favor for pre-trained models (or fine-tuning them) when a locally trained representation from scratch for the GCN underperforms the pre-trained or fine-tuned models.
> >
> > W2: Thanks for the explanation on the limitations. Given the limitations of the duration in the peer review process, I think demonstrating LORAX + Uni-Mol is sufficient. I still stand by my initial stance that the authors should have dedicated more effort towards benchmarking more recent foundation models, especially since the community would naturally gravitate towards using recent models than older ones, making hard to judge whether the fine-tuning process is also beneficial for larger models, or if the observed performance gains are only applicable to smaller models.
> >
> > W3: I'm curious why the error bars are much wider for the importance of the token in the new figure (Figure 1 bottom) compared to the previous Figure 4B? And to be clear, is the importance extracted from each of the five cross-validated models?
> >
> > W4: In the introduction, it is stated that the results are derived from odorant-receptor binding tasks -- but this is not stated in the abstract. The outlook does hint towards the generalizability of LORAX towards other olfactory datasets, but I would much prefer that it is stated earlier and more explicitly in the discussion that LORAX has only been shown for olfactory binding tasks. The first sentence of the discussion also makes no mention to the odorant-receptor binding task, and the claims in the first paragraph of the discussion is particularly problematic in this aspect as these conclusions have only been shown specifically for odorant-receptor binding tasks, and not generally for all olfactory tasks. I may have missed other parts of the work where this was an issue for me before, but these stuck out to me in the current re-read of the work.
> >
> > W5: In line ~311 there was a reference to a "raw" representation -- I think this confused me as I was not sure if you were allowing the molecular and protein representations to attend to each other or if you were using the representations straight out of their respective models.
> >
> > Q1: Acknowledged -- some level of exploratory data analysis in the Appendix for each of the datasets would be appreciated for other readers who are not familiar with the datasets.
> >
> > Q2: A small description of the derivation of the neural representation would be useful in the Appendix as well.

---

> > > ### Author Response · Authors · 2025-11-17
> > > **Response to JDeV's Official Comment**
> > >
> > > We appreciate the reviewers' timely response and the clarifications!
> > >
> > > # W1
> > >
> > > Thanks to the reviewer for clarifying this point! We will work on both a GCN comparison for Table 1 and Appendix E, as well as reimplementing MolOR and the work from Hladiš et al. to get a better overall picture for Table 3. We would like to note that we have some other requests, namely a hyperparameter sweep from reviewer r6QY, that are taking a decent amount of GPU resources at the moment. We will be sure to make time for this, although for full transparency, we are mildly hardware limited at the moment as these models are quite large and take a while (~48hrs) to train.
> > >
> > > # W2
> > >
> > > We will work to incorporate Uni-Mol into LORAX and update you when we have done so! We again want to note that we are hardware limited at the moment, but will do our best to incorporate this before the end of the discussion period.
> > >
> > > # W3
> > >
> > > The error bars are from 5-fold cross validation, yes. Thanks for noting this, we have added it to the figure caption in the supplement.
> > >
> > > It is interesting that there is more variability in the weighting for the LORAX model than there was in Figure 4B. In some folds, the weight on the LORAX [cls] token is very high (e.g., 99% [cls] favored), whereas in other folds it is much lower (e.g., fold 4 for LORAX was only 15% [cls] favored). In contrast, the ProSmith models generally attribute larger weights to the original protein and chemical foundation model representations, with the highest [cls] token weighting being 54% across all folds and models. Notably, this was the only fold (1 in 70) where the XGBoost ensemble using the ProSmith model weighted its' [cls] token higher than the original chemical and protein foundation model.
> > >
> > > The increased variance can likely be attributed to differences across training data within the folds, as they are randomly split and may favor the [cls] representation in some cases and in others not. Despite this variability, on average, the ensemble appears to weight the LORAX [cls] token higher.
> > >
> > > # W4
> > >
> > > We have updated the text in the areas the reviewer noted. More specifically we noted that the work that we have done is specifically for odorant-olfactory receptor binding tasks in both the abstract and the first paragraph in the discussion. We would like to note, however, that even in the space outside of odorant-receptor binding, there has not, to our knowledge, been a comprehensive evaluation of multiple physicochemical descriptors and their relation to foundation models, nor has anyone utilized LoRA to fine tune foundation models for any olfactory task. We will continue to go through the text and make sure we are not over stating claims about the utility of the model. If the reviewer still believes the abstract and the 1st paragraph of the discussion to be an overestimate of our claims, please let us know!
> > >
> > > # W5
> > >
> > > The cross attention block in the LORAX model allows for the chemical and protein representation to attend to each other. This means the [cls] token contains information from this cross attention. Line ~311 is highlighting what input features are going into the 3 XGBoost models in the ensemble prior to the transformer model. We concatenated these representations because 1) it makes our method directly comparable to ProSmith, and 2) the [cls] token should contain sufficient information about the combination of the odorant and receptor to not need another cross-attention layer in the ensemble. We have updated the explanation of the XGBoost ensemble in the paper to make it more clear. Let us know if there is anything else we can do here!
> > >
> > > # Q1
> > >
> > > We agree with the reviewer here, we have added a small table in the Appendix (Table 11), highlighting some of the features of the various datasets.
> > >
> > > # Q2
> > >
> > > We thank the reviewer for noting this and have added a short description of how we calculated 'neural_representation' in the appendix (Appendix D, Neural representation generation).

---

> ### Author Response · Authors · 2025-11-19
> **Completed GCN Experiment and Added to Manuscript**
>
> We would like to update the reviewer with our progress on the GCN experiment the reviewer recommended. We have trained a GCN model on all the datasets in section 3 and have reported the results in Appendix C of our paper. Please see the appendix for details on our experiemnt! With these new results we can confirm, with stronger reliability, that molecular features alone are insufficient to perform prediction. We appreciate the reviewer for recommending this experiement and believe that it adds valuable insight, and further corroborates our main results in the paper.

---

> ### Author Response · Authors · 2025-11-24
> **Update On Follow-up Experiments**
>
> Hello Reviewer jDeV,
>
> We would like to give an update on our progress for the follow up experiments the reviewer has reuested.
>
> ## Updating MolOR in Table 3
> We have implemented the MolOR model, and we are running it on the M2OR dataset to generate the MolOR (weighted) row for Table 3. Training this model is a substantial undertaking, requiring roughly 100 hours of compute. We are prioritizing this experiment and are optimistic that results will be available before the rebuttal deadline!
>
> ## Incorporating UniMol
> We have also completed a major update to the LORAX architecture, enabling it to use UniMol as a chemical foundation model. Training on the M2OR dataset is currently in progress (expected training time: ~60 hours), and we anticipate sharing results during the discussion period.
>
> ## Updating  Hladiš et al. in Table 3
> As the MolOR model has more missing datapoints in Table 3, we felt it was most impactful to begin there and address those gaps first. Due to the effort and compute required to integrate and train MolOR as well as LORAX with UniMol, retraining the model from Hladiš et al. is likely beyond our current resources for this review cycle. Nonetheless, if time permits, we will try to include it and will share any findings as they become available.
>
> We appreciate the reviewers feedback, which continues to help us improve our work, and look forward to presenting these new results as soon as they are ready!

---

> > ### Comment · Reviewer_jDeV · 2025-11-24
> >
> > Thanks for your continued engagement and keeping me updated on your progress. I acknowledge the comments and clarifications provided in your responses above.
> >
> > On the GCN model -- were similar results generated for the ProSmith models? I see that the results are only reported for the MO models.

---

> > > ### Author Response · Authors · 2025-11-25
> > > **Other GCN Experiments**
> > >
> > > We belive there may have been a slight misunderstnading of the reviewers initial comment on our part! When the reviewer requested a "supervised GCN baseline," we interpreted this as a GCN trained $\text{\textit{only}}$ on molecular data, without incorporating protein information (i.e. as an MO model). Our rationale was that directly incorporating a GCN into the ProSmith framework is nontrivial, as ProSmith is inherently designed around combining foundation models rather than graph neural networks, and its integration with GCNs is not straightforward. Additionally, adding a GCN to the MP model would alter its intended purpose as a linear model combining molecular and protein features.
> > >
> > > We would like to note that the MolOR architecture combines a molecular GCN with a protein foundation model, and we are currently retraining this model for Table 3. This model is conceptually very similar to what would be expected if ProSmith were extended with a GCN. For example, both methods combine representations through attention, MolOR using a cross attention block, and ProSmith concatentating both molecular and protein representation and doing self attention (effectively doing the same thing).
> > >
> > > Given these considerations, would the reviewer be satisfied with our results being presented in Table 3, where MolOR and ProSmith are directly compared? If the reviewer would still prefer that we adapt ProSmith with a GCN, we are open to pursuing this direction based on the reviewers guidance. However, we would like to note that implementing this change may require additional time and is not trivial.

---

> ### Author Response · Authors · 2025-11-28
> **Completed MolOR Retraining**
>
> We would like to update the reviewer on our progress retraining the MolOR model for Table 3. We have successfully retrained MolOR and have reported all relevant metrics in the latest version of the paper. The inclusion of updated MolOR results provides even stronger evidence for the advantages of LORAX over existing SOTA models, as we observe substantial performance improvements with LORAX. These results further support our conclusion that fine-tuning chemical foundation models with LORAX yields significant improvements compared to approaches that use GNNs alongside protein foundation models. We thank the reviewer for suggesting this experiment, which we believe has strengthened our work and underscores the importance of the LORAX approach!

---

> > ### Author Response · Authors · 2025-12-01
> > **Completed training LORAX with UniMol**
> >
> > We would like to update the reviewer and let them know that we have completed retraining LORAX with the Uni-Mol2-84M model (Ji et al., 2024) and have added the results to the supplement in Table 8. Our analysis shows that integrating Uni-Mol2-84M did not improve predictive performance for LORAX on the M2OR dataset. While LORAX with Uni-Mol2-84M achieved slightly higher precision, LORAX with the ChemBERTa-77M-MTR model (used in the main text, Table 3) consistently outperformed Uni-Mol2-84M across all metrics.
> >
> > This result is interesting, since Uni-Mol2-84M is a newer and more complex model compared to ChemBERTa-77M-MTR (which is based on the RoBERTa architecture). One possible explanation could be that ChemBERTa-77M-MTR’s pretraining objective or data are more closely aligned with the properties of the M2OR dataset, while Uni-Mol2-84M’s broader focus, including 3D molecular representations, may not as important for this particular task. That being said, we view this as an exciting direction for future research and intend to explore larger Uni-Mol2 models. We thank the reviewer for this suggestion and we hope we have fully addressed their concerns!

---

### Meta-Review · Area_Chair_Txqj · 2026-01-08

**Summary:**

This paper proposes LORAX, a Low-Rank Adaptation (LoRA) method for fine-tuning chemical foundation models. The approach focuses on predicting odorant-receptor binding. The authors demonstrate that the chemical foundation models align well with physicochemical descriptors.

**Reviewer Concerns:**

The reviewers raised the following concerns

1. Technical novelty (P2Me): the reviewer mentioned that the approach is incremental because it is a combination of ProSmith + LoRA using ChemBERTa.
2. Comparison with recent approaches (jDeV, Cjx2, r6QY): the reviewer requested a comparison with UniMol (newer foundation models), GCNs, and MolOR
3. Hyperparameter analysis (r6QY, Cjx2): the reviewer requested abilation study with LoRA parameters. The reviewer requested statistical tests to validate improvements

**Reviewer Scores:**

The reviewer initially gave 6(jDev), 4(Cjx2), 4(r6QY), 2(P2Me). Due to the OpenReview issue, the newly assigned AC could not track the updated score, but confirmed that reviewers Cjx2 and r6QY have mentioned the increase in their initial scores.

Here is the summary of how the authors provided a rebuttal regarding concerns raised above. Regarding concern 1, the authors argue that applying LoRA to chemical foundation models for olfaction is novel, which tackles specific domain gaps. Regarding 2, the authors integrated and tested LORAX with UniMol and reported results. The authors also retrained MolOR and GCN baselines. Regarding 3, the authors performed a comprehensive sweep of ranks and adapter modules. The authors also conducted Friedman tests with FDR correction, and they provided detailed results.

Overall, AC confirms that the authors of the paper properly addressed the major concerns raised by reviewers. AC recommends the paper's acceptance. However, AC notes weak confidence, since the major concern raised by P2Me has valid points, especially technical novelty. AC's decision is largely based on the value of the new specific problems this paper addresses, even though the proposed approach is built on well-established work.

---

### Decision · Program_Chairs · 2026-01-26

Accept (Poster)